# Coherent hexagonal platinum skin on nickel nanocrystals for enhanced hydrogen evolution activity

Kai Liu[1,5], Hao Yang [2,5], Yilan Jiang [3,4,5], Zhaojun Liu [1], Shumeng Zhang[1], Zhixue Zhang[1], Zhun Qiao[1], Yiming Lu[2], Tao Cheng [2]✉, Osamu Terasaki [3,4], Qing Zhang[3,4]✉ & Chuanbo Gao [1]✉

Metastable noble metal nanocrystals may exhibit distinctive catalytic properties to address the sluggish kinetics of many important processes, including the hydrogen evolution reaction under alkaline conditions for water-electrolysis hydrogen production. However, the exploration of metastable noble metal nanocrystals is still in its infancy and suffers from a lack of sufficient synthesis and electronic engineering strategies to fully stimulate their potential in catalysis. In this paper, we report a synthesis of metastable hexagonal Pt nanostructures by coherent growth on $3d$ transition metal nanocrystals such as Ni without involving galvanic replacement reaction, which expands the frontier of the phase-replication synthesis. Unlike noble metal substrates, the $3d$ transition metal substrate owns more crystal phases and lower cost and endows the hexagonal Pt skin with substantial compressive strains and programmable charge density, making the electronic properties particularly preferred for the alkaline hydrogen evolution reaction. The energy barriers are greatly reduced, pushing the activity to 133 mA $cm_{geo}^{-2}$ and 17.4 mA $\mu g_{Pt}^{-1}$ at −70 mV with 1.5 μg of Pt in 1 M KOH. Our strategy paves the way for metastable noble metal catalysts with tailored electronic properties for highly efficient and cost-effective energy conversion.

The electrocatalytic hydrogen evolution reaction (HER) has attracted great attention because it is associated with the water-splitting production of H₂, a clean alternative to traditional fossil fuels[1–5]. So far, Pt is widely recognized as one of the best-performing catalysts for the HER[6,7]. However, most Pt catalysts essentially share the same face-centered cubic (*fcc*) phase as the bulk material. Recent studies suggest that metastable-phase noble metal nanocrystals often show distinctive catalytic properties[8–16]. For example, Pt-Ni alloy nanocrystals with a

metastable hexagonal close-packed (*hcp*) phase exhibit improved HER activity in alkaline media[17,18]. However, the catalytic activities are still not comparable to those of state-of-the-art Pt catalysts with a conventional *fcc* phase for this reaction[19,20]. From an energetic perspective, the binding energy of hydrogen ($\Delta G_{H^*}$) should be close to 0 for efficient hydrogen desorption from the catalyst[21,22]. The $\Delta G_{H^*}$ value has not reached ideal levels and is often too high for most Pt-based catalysts, which we believe holds true for metastable ones. To this end, the

[1]State Key Laboratory of Multiphase Flow in Power Engineering, Frontier Institute of Science and Technology, Xi'an Jiaotong University, Xi'an, Shaanxi 710054, China. [2]Institute of Functional Nano & Soft Materials (FUNSOM), Jiangsu Key Laboratory for Carbon-Based Functional Materials & Devices, Joint International Research Laboratory of Carbon-Based Functional Materials and Devices, Soochow University, Suzhou, Jiangsu 215123, China. [3]Center for High-resolution Electron Microscopy (ChEM), School of Physical Science and Technology, ShanghaiTech University, Shanghai 201210, China. [4]Shanghai Key Laboratory of High-resolution Electron Microscopy, ShanghaiTech University, Shanghai 201210, China. [5]These authors contributed equally: Kai Liu, Hao Yang and Yilan Jiang. ✉e-mail: tcheng@suda.edu.cn; zhangqing1@shanghaitech.edu.cn; gaochuanbo@mail.xjtu.edu.cn

electronic structure of metastable Pt-based catalysts should be modulated toward a lower *d*-band center position relative to the Fermi level according to the *d*-band center theory[23–25]. Therefore, we believe that a metastable Pt phase with modulated electronic structure may show significantly improved catalytic activity in alkaline HER.

In this context, a coherent metastable Pt phase of a few atomic layers on Ni and potentially other 3*d* transition metal nanocrystals represents an ideal catalyst model for the alkaline HER. First, Ni and other transition metal nanocrystals are competent substrates to guide the formation of metastable noble metal phases. To date, limited success has been achieved in the direct wet-chemical synthesis of metastable-phase noble metal nanocrystals[26–28]. Alternatively, they can be obtained by coherent crystal growth on metal substrates that already possess the specific phase. However, such substrates, including hexagonal Au[29,30] and Pd[31,32], are not readily available by conventional syntheses. In contrast, 3*d* transition metals possess more diverse crystal phases. For example, both *fcc* and *hcp* phases can be found in Ni, which may enrich the synthesis of metastable-phase noble metal nanocrystals by coherent crystal growth. Second, Ni and other 3*d* transition metals are unique electronic modulators for the coherent noble metal phases. On the one hand, their atomic radii are typically smaller than those of Pt and many other noble metals, which may induce strong compressive strains in the coherent noble metal phase, broaden its *d* band, and downshift the *d*-band center position[33–35]. On the other hand, their electronegativities are generally lower than those of common noble metals, which allows significant electron transfer from the 3*d* transition metal substrate to the coherent noble metal phase, leading to an upshift of the Fermi Level in the noble metal phase. Both can significantly weaken $\Delta G_{H^*}$ for accelerating the HER, which cannot be achieved by other common substrates such as Au and Pd in previous syntheses. Third, the structure of few-layer Pt on a Ni or other 3*d* transition metal substrate allows maximal exposure of Pt atoms on the surface, not buried underneath, for ready accessibility. This catalyst design greatly reduces the overall cost thanks to the high abundance of 3*d* transition metals in the earth's crust. Despite these merits, such a synthesis has not been established so far for metastable-phase noble metal catalysts, even for general noble metal ones. This is because a galvanic replacement reaction readily occurs between 3*d* transition metal substrates and noble metal salts, causing destruction of the substrates and therefore a loss of well-defined nanostructure[36–38].

Herein, we report the synthesis of a coherent metastable *hcp*-Pt phase on *hcp*-Ni nanocrystals by overcoming the galvanic replacement reaction and reveal its unique electronic properties and significantly improved catalytic activities in the alkaline HER (Fig. 1). We first synthesized branched Ni nanocrystals with an *hcp* phase. The key to the coherent growth of Pt on these *hcp*-Ni nanobranches is to introduce oleylamine as a strong ligand, which significantly lowers the reduction potential of the Pt salt and thus its tendency to react with the Ni nanobranches by galvanic replacement. While Pt retains its *fcc* phase and intrinsic lattice size on (0001) facets of the Ni nanobranches, it successfully replicates the *hcp* phase of the Ni nanobranches on non-(0001) facets with the $d_{0002}$ spacing perpendicular to the (0001) planes shrunk to that of the Ni substrate, showing strong compressive strains in the *hcp*-Pt skins. The well-defined core-shell nanostructure further allows layer-by-layer electron transfer from the Ni core to the Pt skin. Therefore, the electronic structure of the *hcp*-Pt skins can be effectively modulated by the Ni cores. Moreover, abundant step sites are formed on the *hcp*-Pt skins, inheriting from the unique branch-like *hcp*-Ni templates. Experiments and density functional theory (DFT) calculations confirm that the metastable *hcp* phase of the Pt skins, the unique core-shell electronic interactions, and the surface step sites significantly lower the energy barriers of the alkaline HER, pushing catalytic activities to a high level (Fig. 1). This work paves an avenue to a class of core-shell structured noble metal catalysts with designable phases, unique electronic properties, and low cost, offering new opportunities in designing highly efficient catalysts for energy conversion.

## Results
### Synthesis and structural analysis of the *hcp*-Ni@Pt-skin nanostructure

A metastable *hcp*-Pt skin was synthesized by coherent crystal growth of Pt on *hcp*-Ni nanocrystals (Fig. 2). To this end, *hcp*-Ni nanobranches were first synthesized by reducing Ni(acac)$_2$ (acac: acetylacetonate) in a solvothermal system. A small amount of H$_2$PtCl$_6$ (Pt/Ni = 2%) was introduced into the synthesis to afford *fcc*-Pt nanoparticles as seeds to trigger the crystal growth. The resulting product shows an X-ray diffraction (XRD) pattern with well-resolved {01–10}, (0002), and {01-11} reflections from a typical *hcp* lattice (2H phase) (Supplementary Fig. 1). Transmission electron microscopy (TEM) images show that the product is composed of nanocrystals with rod-like branches extending along the [0001] direction (Supplementary Fig. 1). The unique morphology of the Ni nanocrystals may arise from the growth of *hcp*-Ni branches on multiple {111} facets of individual polycrystalline Pt seeds[39]. Depending on the seed structure, the number of *hcp*-Ni branches in each nanocrystal varies from 4 to 9 (Supplementary Fig. 2).

The coherent growth of an ultrathin Pt skin on the Ni nanocrystals is challenging because a galvanic replacement reaction quickly occurs between the Ni nanocrystals and the Pt salt upon mixing in the

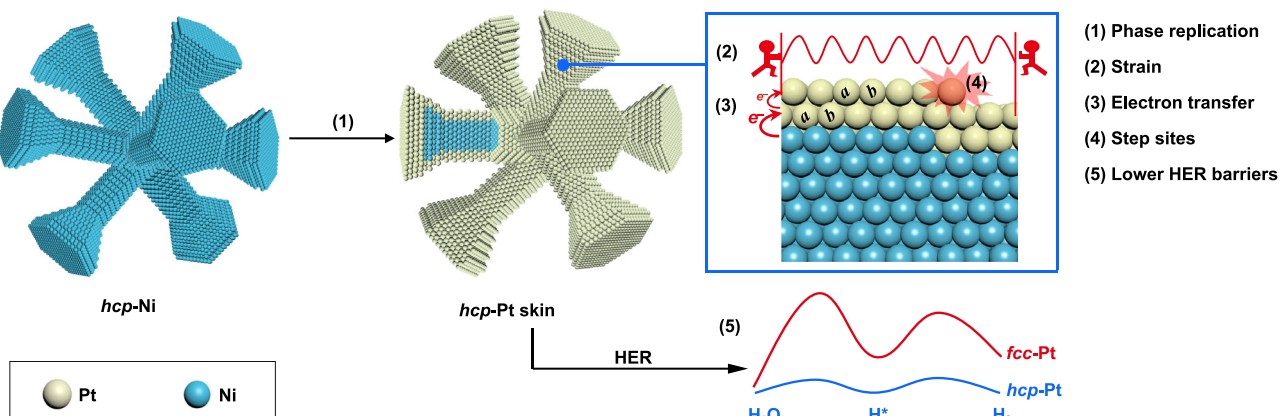

**Fig. 1 | Synthesis and electronic interactions of a coherent metastable *hcp*-Pt skin on an *hcp*-Ni nanocrystal for the HER.** A coherent Pt skin is grown on an *hcp*-Ni nanocrystal by suppressing the galvanic replacement reaction, leading to a phase replication, strong compressive strains, layer-by-layer Ni–Pt electron transfer, and formation of surface step sites, which lower the energy barriers of the HER for improved catalytic activity.

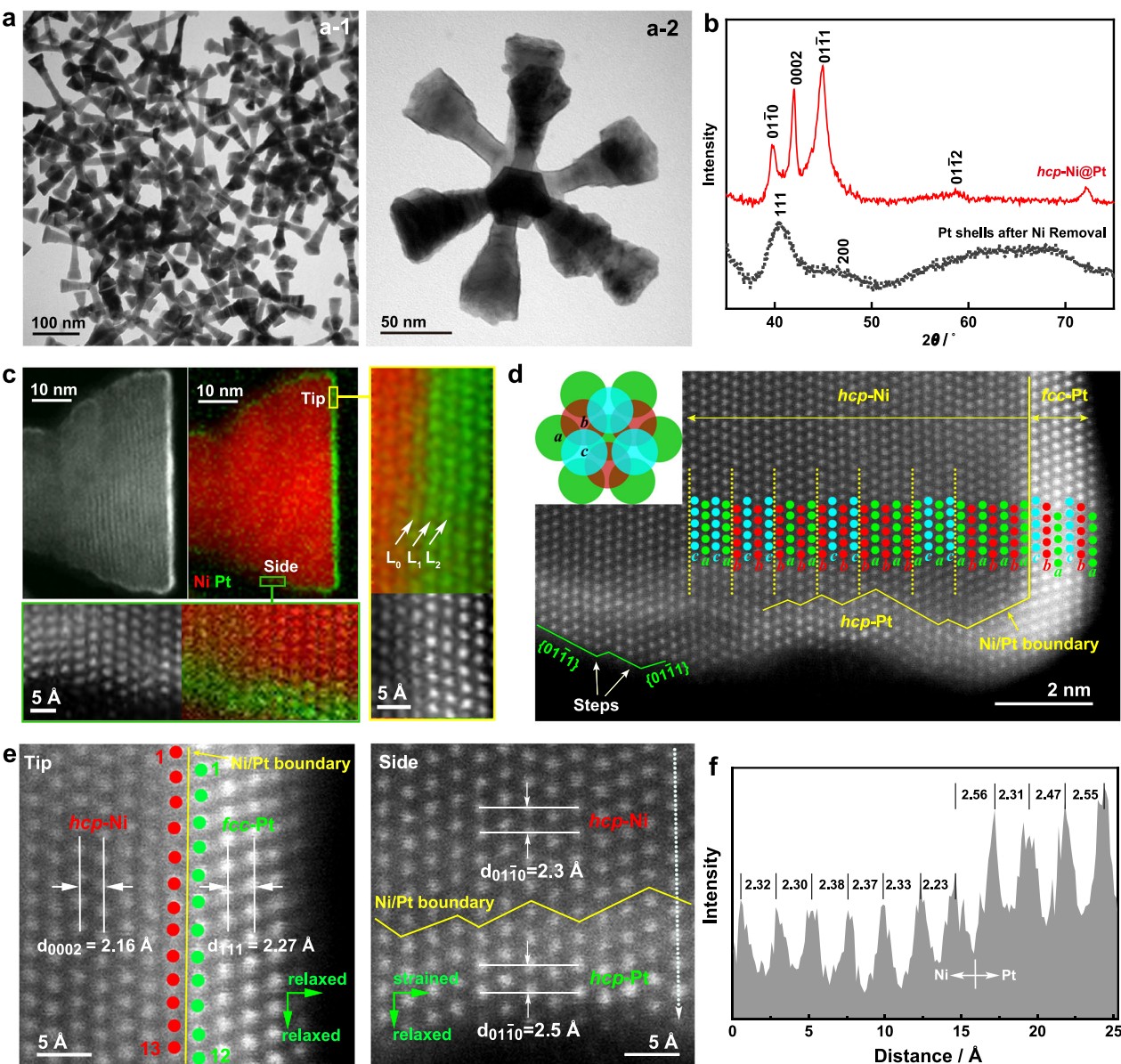

**Fig. 2 | Structural and phase analysis of the *hcp*-Ni@Pt-skin core-shell nanobranches. a** Low-magnification TEM images. **b** XRD patterns of the *hcp*-Ni@Pt core-shell nanobranches and Pt shells after etching of Ni. **c**–**f** $C_s$-corrected HR-STEM analysis. **c** Atomic-resolution EELS mappings acquired at the tip and side of the nanobranches with corresponding HAADF-STEM images. **d** Phase analysis of the nanobranches, showing an *hcp* structure of both the Ni core and the Pt skin at the side and an *fcc* structure of the Pt skin at the tip. Inset: Positions of *a*, *b*, and *c* in a close-packed structure. The surface step sites are also indicated in the image. **e** Atomic resolution HAADF-STEM images, showing the lattice spacing correlation between the Ni substrate and the Pt skin at the tip (left) and side of the nanobranch (right). **f** Intensity profile along the dashed arrow in (**e**), showing the lattice spacings along the <01–10> direction at the side of the nanobranches.

synthesis. A careful literature search suggests that Ni@Pt core-shell nanostructures have been reported previously[40–47]. However, these nanostructures were usually synthesized by sequential reduction of Ni and Pt without delicate design for suppressing the galvanic replacement side-reaction, which may lead to destruction of the Ni/Pt boundary, significant Ni-Pt atomic mixing, and loss of control over the structure. As a result, many of these prior works failed to provide unambiguous evidence to prove a well-defined core-shell structure as a heterojunction of monometallic Ni and Pt with sufficient control. To overcome the galvanic replacement reaction, we previously developed a strategy that utilizes strong ligands to reduce the reduction potential of the noble metal salt[48–52]. This strategy is particularly successful in growing Au, Pt, and Pd on less-stable Ag nanocrystal substrates. When non-noble transition metal nanocrystals are used as the substrate, the synthesis becomes difficult, especially when strict control over crystal

growth is required[53]. To address this challenge, we here advance this strategy by introducing oleylamine as a strong ligand to substantially decrease the reduction potential of $H_2PtCl_6$ so that the galvanic replacement reaction becomes thermodynamically unfavorable. Moreover, the synthesis is carried out at a high temperature (144 °C) with a low injection rate of Pt precursor into the synthesis system, which allows sufficient diffusion of Pt atoms across Ni surface to form a complete shell, despite the large Ni-Pt lattice mismatch[54–56]. By this means, the coherent growth of a Pt skin on the Ni nanobranches has been successfully achieved without involving the galvanic replacement reaction (Fig. 2a, Supplementary Fig. 3; control experiments, Supplementary Figs. 4 and 5).

The XRD pattern of the Ni@Pt core-shell nanobranches shows characteristics of a typical *hcp* phase (Fig. 2b). The core-shell structure can be confirmed by electron energy loss spectroscopy (EELS)

mapping with atomic resolution captured on a spherical-aberration ($C_s$)-corrected high-resolution scanning transmission electron microscope (HR-STEM), which clearly shows a Pt skin at the outer rims of the Ni nanocrystal (Fig. 2c). The high-angle annular dark-field scanning transmission electron microscopy (HAADF-STEM) images show atomic-number dependent contrast, higher at the rim and lower in the core, corresponding to Pt and Ni atoms, respectively, consistent with the EELS elemental mapping results. A closer inspection of the tip and side of the nanobranch suggests that the skin is composed of Pt with negligible Pt-Ni mixing across their interface (observable only in the first layer as indicated by $L_0$ in Fig. 2c), different from common coherent crystal growth that usually causes significant interfacial alloying[57]. A previously reported Pt-Ni phase diagram shows that Ni and Pt can dissolve into each other to reach a concentration of 5 and 8%, respectively, at the reaction temperature of 144 °C[58]. However, it requires a long time to reach equilibrium. Therefore, although there is a tendency for bimetallic interdiffusion across the interface, it has been largely suppressed in our system, which can be partially attributed to the core-shell lattice mismatch[59]. From the perspective of energetics, Ni and Pt possess very similar surface energies (Ni: 2.45 J m$^{-2}$; Pt: 2.48 J m$^{-2}$), which may have made outward diffusion of Ni less favorable during the coherent crystal growth[60]. The negligible Pt-Ni mixing offers an opportunity to finely tune the Ni–Pt electron transfer and therefore the electronic property of the Pt skin via its thickness.

To understand how the Pt skin replicates the phase and structure of the Ni substrate, the atomic arrangements at the tip and side of the core-shell nanobranches were investigated by atomic-resolution $C_s$-corrected HAADF-STEM imaging (Fig. 2d). The Ni atoms (low-contrast spots) align in an *abab* sequence (equivalent to *acac* or *bcbc* sequences) along the growth direction of the nanobranch, which is a typical packing structure of the *hcp* phase, albeit with abundant stacking faults. At the tip of the nanobranch, Pt atoms (high-contrast spots) were observed in an *abcabc* stacking sequence, which is the packing structure of the *fcc* phase. Therefore, on (0001) facets of the *hcp*-Ni, the Pt skin does not replicate the phase of Ni substrate but takes its intrinsically stable *fcc* phase, growing along <111> directions. On non-(0001) side facets of the Ni branch, the Pt atoms align in the same *abab* sequence as the Ni atoms in the core, suggesting successful replication of the *hcp* phase of the Ni core by the Pt skin. Phase replication thus exhibits obvious anisotropy on different facets of the substrate. It is worth noting that the ultrathin *hcp*-Pt phase is stable only when it is supported on Ni substrate. Upon the substrate removal, it readily transforms from the *hcp* phase into an *fcc* phase, confirming the intrinsic instability of the *hcp*-Pt phase under ambient conditions (Fig. 2b). A close inspection of the surface structure suggests that the *hcp*-Pt skin on the side exposes {01-11} facets with abundant step sites, inheriting from the Ni substrate (Fig. 2d). These atomic steps may serve as extra active sites, favorable for improving the catalytic properties[61].

We further examined the effect of the Ni substrate on the lattice spacing of the Pt skin (Fig. 2e). At the tip of the branch, the $d_{111}$ spacing perpendicular to the {111} plane in the *fcc*-Pt skin is 2.27 Å, which is ~5% larger than the $d_{0002}$ spacing in the *hcp*-Ni phase (2.16 Å). Along the Ni/Pt interface, the Pt atoms are not aligned perfectly one-to-one with the Ni atoms, showing obvious mismatch in the atomic alignments. These observations suggest that the *fcc*-Pt skin relaxes to its intrinsic spacing when it is grown on (0001) facets of the *hcp*-Ni substrate. On non-(0001) side facets of the nanobranch, the $d_{01-10}$ spacing changes from 2.3 Å in the *hcp*-Ni phase to 2.5 Å in the *hcp*-Pt phase. Figure 2f shows the intensity profile along the dashed arrow in Fig. 2e. The spacings between the Pt atoms (high-intensity peaks) are statistically larger than those between the Ni atoms (low-intensity peaks), again displaying a tendency of the lattice relaxation in the *hcp*-Pt phase. However, the $d_{0002}$ spacing of the *hcp*-Pt phase is precisely the same as that of the *hcp*-Ni phase (2.16 Å), showing a substantial shrinkage of the lattice spacing in the [0001] direction. The *hcp*-Pt skin thus possesses a compressed hexagonal unit cell in the *c* direction ($a = b = 2.89$ Å, $c = 4.32$ Å; $c/a = 1.49 < 1.633$ for ideal hexagonal sphere packing). Combining these observations, we conclude that the lattice size of the Pt skin is anisotropically affected by the Ni substrate. While the Pt skin tends to relax to its intrinsic spacing, the *hcp*-Pt skins on non-(0001) facets of the nanobranches are significantly compressed by the Ni substrate uniaxially in the [0001] direction. We expect the compressive strain decreases $\Delta G_{H^*}$ by downshifting the *d*-band center of the *hcp*-Pt skin, which is desired for improving the HER activity[33–35].

**Thickness-dependent electronic properties of the *hcp*-Pt skins**
Thanks to the effective suppression of the galvanic replacement reaction, the thickness of the coherent *hcp*-Pt skin can be precisely controlled by simply adjusting the amount of the Pt precursor in the synthesis (Fig. 3, and Supplementary Fig. 6). The resulting core-shell nanobranches are denoted as *hcp*-Ni@Pt$_{nL}$, where *n* represents the average atomic layers in the *hcp*-Pt skin. Figure 3a shows the $C_s$-corrected HAADF-STEM images of the side edges of the *hcp*-Ni@Pt$_{nL}$ ($n = 2.6$, 3.2, and 4.0) nanobranches. The *n*-values were counted statistically from the HR-STEM images, which are found to be linearly related to the fraction of Pt in the nanobranches measured by inductively coupled plasma mass spectrometry (ICP-MS) (Supplementary Fig. 7). It is worth noting that *hcp*-Pt with even fewer atomic layers can also be obtained (*hcp*-Ni@Pt$_{1.3L}$, estimated by ICP-MS, Supplementary Fig. 7). However, the ultralow loading of Pt makes the Ni substrate unstable and prone to oxidation in ambient air, forming a thick NiO layer (~3 nm) (Supplementary Fig. 8), which may cause unexpected inferences to the electronic properties of the Pt skins.

The electronic properties of the *hcp*-Pt skin in the Ni@Pt$_{nL}$ ($n = 2.6$, 3.2, and 4.0) core-shell nanobranches were examined by X-ray photoelectron spectroscopy (XPS) (Fig. 3b). The Pt 4*f* spectra are fitted by two sets of spin-orbit split $4f_{7/2}$ and $4f_{5/2}$ components, corresponding to $Pt^0$ and $Pt^{2+}$, respectively. The peak of $Pt^{2+}$ can be attributed to the partial oxidation of the surface Pt atoms and may also contain the contribution from the plasmon-energy loss. With decreasing thickness of the Pt skins from $n = 4.0$ to 2.6, the $Pt^0$ peak shows a continuous shift in the binding energy from 70.93 to 70.69 eV, suggesting an increase in the electron density in the Pt skins. This observation can be rationalized by a progressive electron transfer from the Ni substrate to the Pt skin due to the differences in work function ($Pt_{111}$, 5.93 eV; $Ni_{111}$, 5.35 eV) and electronegativity (Pt, 2.2; Ni, 1.91)[62], which leads to an upshift in the Fermi level position of the Pt skin and therefore a decreased $\Delta G_{H^*}$ according to the *d*-band center theory. Because electron transfer is a short-range interaction that is pronounced between neighboring atoms (electronic "ligand effect") and significantly weakens to a third atom[63], the electron density of the surface Pt atoms decreases continuously with increasing thickness of the *hcp*-Pt skins. By this means, the electronic property of the *hcp*-Pt skin can be precisely tuned, leading to programmable catalytic properties. Such precise control of electronic property of Pt could not be achieved without suppression of Pt-Ni alloying in the Pt skins. In case of core-shell structure with significant atomic interdiffusion, the degree of Ni–Pt electron transfer becomes unpredictable due to random coordination environments of the metal atoms, leading to a loss of precise control over the electronic property of the Pt atoms in the shells.

**Electrocatalytic performance**
Considering the metastable phase and the unique core-shell interactions, we expect the coherent *hcp*-Pt skins on the *hcp*-Ni substrates may show high activities in the alkaline HER. To verify it, we investigated the activities of the *hcp*-Ni@Pt$_{nL}$ catalysts compared to *fcc*-Ni@Pt-skin and the commercial Pt/C. The *fcc*-Ni@Pt-skin catalyst was obtained by phase transformation of the *hcp*-Ni@Pt-skin by thermal annealing without obvious changes in the structure and morphology

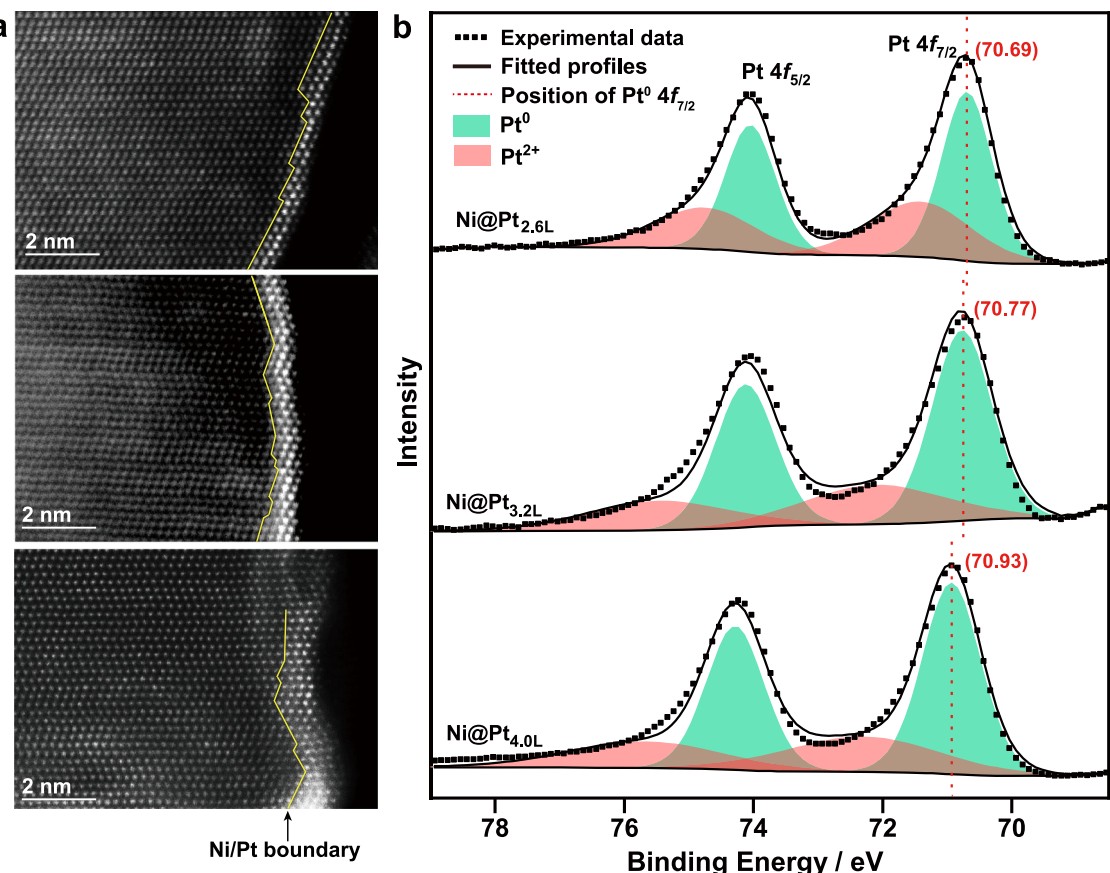

**Fig. 3 | Thickness-dependent electronic properties of the coherent *hcp*-Pt skins on Ni substrates. a** HAADF-STEM images of the *hcp*-Ni@Pt₂.₆ₗ (top), *hcp*-Ni@Pt₃.₂ₗ (middle), and *hcp*-Ni@Pt₄.₀ₗ (bottom) nanobranches. **b** Pt 4*f* XPS spectra of the *hcp*-Ni@Pt$_{nL}$ nanobranches fitted by 4*f*$_{7/2}$ and 4*f*$_{5/2}$ components.

(Supplementary Fig. 9). All the nanocrystals were supported on carbon nanotubes and transferred to a rotating disk electrode (RDE, 0.196 cm²) with a fixed Pt mass of 1.5 μg. The HER activities of the catalysts were evaluated by linear sweep voltammetry (LSV, 90% *iR* compensation) in N₂-saturated 1 M KOH in the potential range of 0 ~ −0.10 V vs. a reversible hydrogen electrode (RHE) at a scan rate of 10 mV s⁻¹, with the current densities normalized to the mass of Pt and the geometric area of the electrode, respectively (Fig. 4a, b; other experimental parameters, Supplementary Figs. 10, 11). The mass activities of the *hcp*-Ni@Pt₁.₃ₗ, *hcp*-Ni@Pt₂.₆ₗ, *hcp*-Ni@Pt₃.₂ₗ, *hcp*-Ni@Pt₄.₀ₗ, *fcc*-Ni@Pt₂.₆ₗ, and the commercial Pt/C are 2.24, 17.4, 12.1, 7.95, 3.26, and 1.11 mA μg$_{Pt}$⁻¹, respectively, at −70 mV (Fig. 4c and Supplementary Fig. 12). Notably, the *hcp*-Ni@Pt₂.₆ₗ catalyst showed the highest mass activity among all shell thicknesses investigated, which is 15.7 times greater than that of the commercial Pt/C, making it comparable to or better than the state-of-the-art Pt-based catalysts reported to date (a survey of reported values with different test conditions, Supplementary Table 1)[7,17–20,64–71]. The catalytic activity of the *hcp*-Ni@Pt₂.₆ₗ catalyst normalized to the geometric area of the electrode reached 133 mA cm$_{geo}$⁻² at −70 mV with 1.5 μg of Pt, which is 15.6 times greater than that of the commercial Pt/C (Fig. 4b). These results validate the design of the metastable catalyst toward improved activities for the alkaline HER.

The metastable *hcp* phase of the Pt skin plays a critical role in the HER. Because both *fcc*- and *hcp*-phase Pt skins exist in the *hcp*-Ni@Pt nanobranches, it is difficult to discriminate the contributions from each component experimentally in the HER. Instead, we compared the activities of the *hcp*-Ni@Pt nanobranches with their *fcc* counterparts obtained by thermal phase transformation (Fig. 4a–c). At −70 mV vs. RHE, the activity values of the *hcp*-Ni@Pt₂.₆ₗ

nanobranches (17.4 mA μg$_{Pt}$⁻¹, 133 mA cm$_{geo}$⁻²) are 5.3 times greater than those of the *fcc* counterparts (3.26 mA μg$_{Pt}$⁻¹, 24.9 mA cm$_{geo}$⁻²), albeit the similar core-shell structure, which unambiguously confirms the critical role of the metastable *hcp* phase in improving the catalytic activity of the Pt skins in the alkaline HER.

The effects of the modulated electronic structure of the Pt skins on the HER activity are also clear. All the Ni@Pt nanobranches, despite the phase and skin thickness, exhibit higher catalytic activities than the monometallic Pt/C (Fig. 4c), which can be attributed to the compressive strain in the Pt skins that lowers the $\Delta G_{H^*}$ for accelerating the HER[68,72]. With decreasing thickness of the Pt skin, the mass activity exhibits a volcano-shaped profile peaking at a thickness of 2.6 L (*hcp*-Ni@Pt₂.₆ₗ) (Fig. 4c). It is difficult to measure the electrocatalytically active surface areas (ECSAs) of the catalysts directly by the underpotential deposition (UPD) of hydrogen because the Ni cores are prone to oxidation at positive potentials. However, given the atomic layer number (*n*) of Pt obtainable statistically by $C_s$-corrected HAADF-STEM imaging, the ECSA of the *hcp*-Ni@Pt$_{nL}${01-11} catalysts can be estimated to be 222.8/*n* m² g⁻¹ by geometrical calculations (Supplementary Fig. 13). By this means, the ECSAs of the *hcp*-Ni@Pt$_{nL}$ (*n* = 1.3, 2.6, 3.2, and 4.0) catalysts were calculated to be 171, 85.7, 69.6, and 55.7 m² g⁻¹, respectively. With these values, the specific activities of the catalysts at −70 mV were estimated to be 1.31, 20.3, 17.3, and 14.3 mA cm$_{Pt}$⁻², respectively (Supplementary Table 2). Alternatively, the specific activity can be evaluated by multiplying the mass activity by the atomic layer number of the Pt skin, which contains fewer assumptions in the calculation (details, Supplementary Information) (Fig. 3d). A similar volcanic trend of the specific activity can be observed with decreasing thickness of the Pt skin. The catalyst with a Pt skin of 2.6 atomic layers shows the highest activity in the alkaline HER. It is reasonable that with

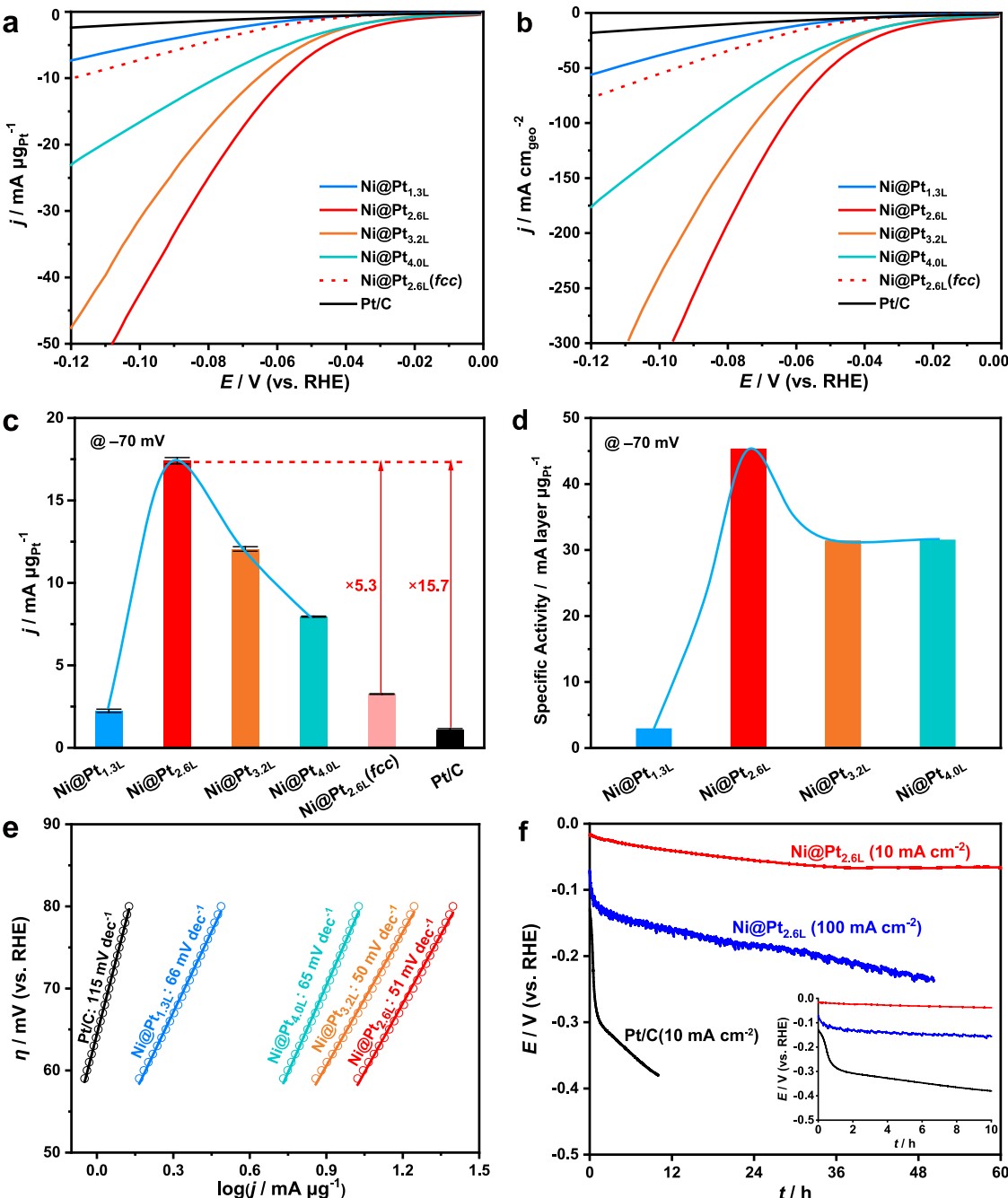

**Fig. 4 | Electrocatalytic performance of the *hcp*-Ni@Pt-skin catalysts in the alkaline HER, compared with *fcc*-Ni@Pt-skin and the commercial Pt/C. a, b** LSV curves of the catalysts with 90% *iR* compensation in $N_2$-saturated 1 M KOH at a scan rate of 10 mV s$^{-1}$. The current densities are normalized to the mass of Pt (1.5 μg) and the geometric area of the electrode (0.196 cm$^2$), respectively. Solution resistances (*R*) were 6.237, 4.114, 5.293, 5.181, 3.604, and 3.399 Ω for measurements with catalysts of *hcp*-Ni@Pt$_{nL}$ (*n* = 1.3, 2.6, 3.2, 4.0), *fcc*-Ni@Pt$_{2.6L}$, and Pt/C, respectively. **c** Mass activities of the catalysts at −70 mV *vs*. RHE, calculated statistically from three parallel measurements (Supplementary Fig. 12), with the error bars indicating the standard deviations. **d** Specific activities (mass activity multiplies atomic layer number of the Pt skin) of the catalysts at −70 mV *vs*. RHE. **e** Tafel plots of the catalysts in the HER. **f** Chronopotentiometric curves of the *hcp*-Ni@Pt$_{2.6L}$ catalyst at constant current densities of 10 and 100 mA cm$_{geo}^{-2}$. The chronopotentiometric curve of the commercial Pt/C was also listed for comparison. Inset: Chronopotentiometric curves in the first 10 h.

decreasing atomic layers of the Pt skin from 4.0 L to 2.6 L, the surface Pt atoms possess increasing electron densities due to the layer-by-layer core-shell electron transfer, which upshifts the Fermi level, decreases $\Delta G_{H^*}$, and thus accelerates the HER. When the thickness of the Pt skin is decreased to 1.3 L, the Pt atoms are directly coordinated to Ni, which may have changed the electron density of states at the Fermi level to afford an unfavorable $\Delta G_{H^*}$.[73,74] We also calculated the exchange current densities with these catalysts, which further shows boosted

reaction kinetics with the core-shell catalysts, compared with the Pt/C (Supplementary Fig. 14). All these results confirm the important role of the delicately engineered electronic properties of the Pt skins on their exceptional catalytic activities in the alkaline HER.

We further employed Tafel analysis to investigate the reaction kinetics of the HER on the different catalysts (Fig. 4e). The Tafel slope with the commercial Pt/C is 115 mV dec$^{-1}$, suggesting the Volmer reaction, *i.e.*, the water dissociation reaction (* + $H_2O$ + e$^-$ → H* + OH$^-$),

is the rate-determining step, consistent with previous results (reference Tafel slope value, ~120 mV dec$^{-1}$)[75,76]. In contrast, the catalysts of *hcp*-Ni@Pt$_{2.6L}$ and *hcp*-Ni@Pt$_{3.2L}$ show significantly lower Tafel slopes of 51 and 50 mV dec$^{-1}$, respectively. These values are very close to the theoretical value of ~40 mV dec$^{-1}$ when the Heyrovsky reaction, i.e., the electrochemical hydrogen desorption reaction (H$^*$ + H$_2$O + e$^-$ → H$_2$ + OH$^-$), becomes the rate-determining step[75]. It suggests that the dissociation of H$_2$O on these catalysts is no longer a huge obstacle for the overall HER. The electrocatalytic hydrogen desorption being the rate-determining step may have substantially accelerated the overall hydrogen evolution rates. It is worth noting that the Tafel slopes with the catalysts of *hcp*-Ni@Pt$_{1.3L}$ and *hcp*-Ni@Pt$_{4.0L}$ are 66 and 65 mV dec$^{-1}$, respectively. These values are in between the typical values of 120 and 40 mV dec$^{-1}$ and are closer to the latter, suggesting that both the Volmer and the Heyrovsky reactions are limiting the overall reaction rate, with the latter being the dominant rate-determining step.

Therefore, we have successfully verified the strong effects of the metastable phase and the modulated electronic structure of the *hcp*-Pt skins on their catalytic activities in the alkaline HER. It is worth noting that the abundant step sites on the *hcp*-Pt surface may have also contributed to the enhanced activity, as these crystallographic defects are well-known hotspots for further lowing the energy barriers in the catalytic reactions[61]. As the defect-free counterparts are unavailable experimentally, the effect of the steps on the catalytic properties of the *hcp*-Ni@Pt-skin catalyst will be discussed later with the aid of DFT calculations.

The *hcp*-Ni@Pt-skin catalyst exhibits improved stability in the alkaline HER compared with the commercial Pt/C (Fig. 4f). In a typical chronopotentiometric test (current density, 10 mA cm$_{geo}^{-2}$), the overpotential (absolute value, and hereafter) rises rapidly from 130 to 380 mV (overpotential increase, 250 mV) in the first 10 h with the commercial Pt/C. At the same current density, the overpotential with the *hcp*-Ni@Pt$_{2.6L}$ nanobranches rises from 17 to 67 mV within 60 h, showing an overpotential increase by only 50 mV in this period, which suggest substantially increased catalyst stability with the catalyst. We also investigated the catalytic stability of the *hcp*-Ni@Pt$_{2.6L}$ nanobranches at a high current density of 100 mA cm$_{geo}^{-2}$. The overpotential showed a slow increase from 80 to 230 mV in 50 h, suggesting satisfactory catalytic stability. TEM imaging reveals that the Pt nanoparticles in the Pt/C catalyst became aggregated (Supplementary Fig. 15) while the *hcp*-Ni@Pt$_{2.6L}$ nanobranches maintained their morphology and structure during the catalysis (Supplementary Fig. 16). The fine structure of the *hcp*-Ni@Pt$_{2.6L}$ nanobranches after catalysis was further inspected by high-resolution TEM and $C_s$-corrected HAADF-STEM (Supplementary Figs. 17 and 18). By carefully examining the crystal structures of the nanobranches at different domains, we can statistically conclude that the metastable *hcp* phase has been largely retained during the long-term electrocatalysis even at the high current density. We also observed minor local areas that transformed into the more stable *fcc* phase, which, however, can only be observed occasionally. Statistically, only ~3% of the *hcp* phase was transformed into the more-stable *fcc* phase on the side facets of the nanobranches. Moreover, the high-resolution electron microscopy images also confirmed that the {01-11} facets and the step sites of the Pt skins were largely retained. The structural integrity of the *hcp*-Ni@Pt core-shell nanobranches contributes to the catalytic durability. Compared with the pristine *hcp*-Ni@Pt$_{2.6L}$ nanobranches, the ones after the catalysis showed a relatively rough surface with less-uniform thicknesses of the Pt skins, suggesting a tendency of the Pt atoms to migrate on the Ni surface to form large ensembles (Supplementary Fig. 18). The surface reconstruction may have altered the electronic structure of the Pt skin to an extent, which contributes to the observed increase in the overpotential during the long-term catalysis. Moreover, a large number of H$_2$ bubbles were observed to form on the electrode surface due to the high current density (Supplementary Fig. 19). The bubbles partially covered the catalyst surface and possibly peeled off the catalyst from the electrode surface, which may also have contributed to the overpotential increase in the catalysis.

## DFT calculations

We carried out DFT calculations to better understand the relationship between the structure of the *hcp*-Ni@Pt-skin nanobranches and their alkaline HER performance (Fig. 5). A model was constructed with two layers of 4 × 4 Pt atoms on top of two layers of 4 × 4 Ni atoms, following an *hcp* arrangement with stepped {01-11} facets. The bottom two layers of the metal atoms were fixed according to the lattice parameters of *hcp*-Ni measured experimentally by electron microscopy (*a*, *b* = 2.65 Å, *c* = 4.32 Å), while all the other atoms were relaxed during the energy minimization. The vacuum layers were set up at least 15 Å to minimize possible interactions between the replicated cells. The atomic coordinates with constraints are listed in Supplementary Information in the VASP format. A model without atomic steps was also constructed to decouple the effect of the metastable phase from that of the surface steps on the catalytic property of the *hcp*-Pt skins.

First, we evaluated the effect of the metastable phase and the surface structure of the *hcp*-Pt skins on their catalytic activities. Figure 5a shows the free energy diagram of water dissociation (Volmer reaction) on *hcp*-Ni@Pt$_{2L}$, *hcp*-Ni@Pt$_{2L}$ with step sites, and *fcc*-Ni@Pt$_{2L}$. The Volmer reaction on *fcc*-Ni@Pt$_{2L}$ needs to overcome the highest energy barrier (1.41 eV). In contrast, the *hcp*-Ni@Pt$_{2L}$ showed a significantly reduced energy barrier (0.84 eV), suggesting that the metastable *hcp* phase played a vital role in reducing the energy barrier of the Volmer reaction. The step sites on the *hcp*-Ni@Pt{01-10} surface further decreased this energy barrier to 0.63 eV, confirming that the step sites are particularly active sites for the Volmer reaction. We also investigated the free energy diagrams of the H$_2$ formation, which can proceed via either the Heyrovsky-type reaction or the Tafel-type reaction. DFT calculation suggests that the Heyrovsky reaction is more favorable (Supplementary Fig. 20). As shown in Fig. 5b, the Heyrovsky reaction on the *hcp*-Ni@Pt$_{2L}$ surface shows an energy barrier of 0.78 eV at −70 mV, which was lower than the energy barrier on the *fcc*-Ni@Pt$_{2L}$ surface (1.0 eV) by 0.22 eV, suggesting that the metastable *hcp* phase also substantially reduced the energy barrier of the Heyrovsky reaction. All the energy barriers obtained from our calculations are close to the reported values in the literature[65,77,78]. By comparing the energy barriers in the full HER pathway (Fig. 5a, b), we can infer that the rate-determining step of the HER was the Volmer reaction on the *fcc*-Ni@Pt$_{2L}$ surface and shifted to the Heyrovsky reaction on the *hcp*-Ni@Pt$_{2L}$ surface, which well agrees with our experimental Tafel analysis. With the latter becoming the rate-determining step, the HER kinetics is substantially accelerated due to the reduced overall energy barrier. It is worth noting that although the step sites on the *hcp*-Ni@Pt$_{2L}$ surface are particular hotspots for the Volmer reaction, the subsequent Heyrovsky reaction at these sites suffers from a higher energy barrier, suggesting that a hydrogen diffusion to the adjacent terrace surface is necessary prior to H$_2$ formation via the Heyrovsky reaction.

With the Heyrovsky reaction becoming the rate-determining step, the reaction kinetics can be understood by using the binding energy of hydrogen ($\Delta G_{H^*}$) as a competent descriptor. This descriptor has been widely used in the study of the HER to simplify the calculation. We examined by DFT calculations the effects of the compressive strains and the core-shell electron transfer on the catalytic activities of the *hcp*-Ni@Pt catalysts in the HER (Fig. 5c, d). As discussed above, the *hcp*-Pt skin was compressively strained by ~5% on non-(0001) facets of the Ni substrates. To verify its effect, we calculated the $\Delta G_{H^*}$ on *hcp*-Ni@Pt surfaces with a range of the compressive strains in the Pt skins (1%, 3%, and 5% relative to the intrinsic lattice size of *hcp*-Pt; the intrinsic lattice size of *hcp*-Pt was derived from *fcc*-Pt as they are both close-packed structures) (Fig. 5c). During the simulations, the positions of the

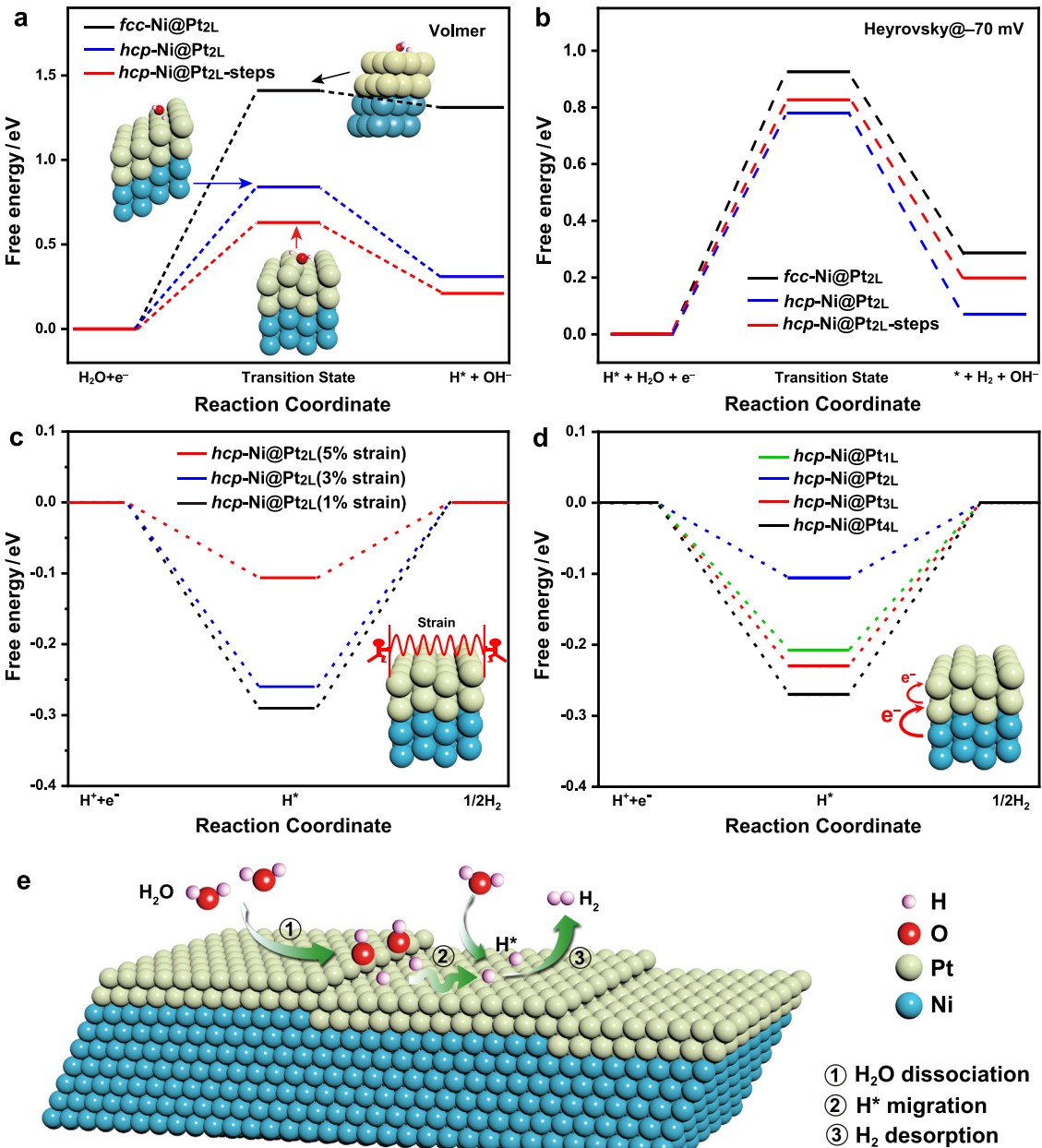

**Fig. 5 | DFT calculation results. a, b** Free energy diagrams of water dissociation (Volmer reaction) and hydrogen desorption (Heyrovsky reaction) at −70 mV on *fcc*-Ni@Pt$_{2L}$, *hcp*-Ni@Pt$_{2L}$, and *hcp*-Ni@Pt$_{2L}$ with step sites, respectively. Inset: Model structures after DFT optimizations. **c** Hydrogen binding energies on *hcp*-Ni@Pt$_{2L}$ with different compressive strains in the Pt kins (1%, 3%, and 5% relative to the intrinsic lattice size of Pt). **d** Hydrogen binding energies on *hcp*-Ni@Pt with different thicknesses of the Pt-skins (1 L, 2 L, 3 L, and 4 L). **e** A cartoon illustrating the HER process on the *hcp*-Ni@Pt surface, including the H$_2$O dissociation at step sites, H* migration to an *hcp*-Pt terrace surface, and the H$_2$ desorption.

bottom two layers of the metal atoms were fixed with various lattice sizes. The specific lattice sizes correspond to different compressive strains (1%, 3%, and 5%) relative to the intrinsic lattice size of Pt. All the other atoms were allowed to undergo relaxation during the energy minimization. The results show that $\Delta G_{H*}$ decreases continuously with increasing strains, consistent with the prediction by the *d*-band center theory, which confirms the contribution of the compressive strain to the accelerated hydrogen desorption and thus the overall rate of the HER. We further calculated the $\Delta G_{H*}$ on *hcp*-Pt skins of different thicknesses to verify the effect of the progressive Ni–Pt electron transfer on their catalytic activities (Fig. 5d). The $\Delta G_{H*}$ drops when the thickness of the Pt skin decreases from 4 L to 2 L on the Ni substrate, in line with the rising electron density. The $\Delta G_{H*}$ further increases when

the thickness of the Pt skin decreases to a monolayer. Of all the models investigated, the *hcp*-Ni@Pt$_{2L}$ catalyst shows the lowest $\Delta G_{H*}$ (−0.11 eV), which well explains the experimental results, i.e., Pt skins with two atomic layers are particularly active for the HER. These results verify the effectiveness of the electronic property modulation of the *hcp*-Pt skins in optimizing their catalytic activities in the alkaline HER.

Upon a comprehensive exploration of all potential full HER reaction mechanisms, DFT calculations indicate that the Volmer-Heyrovsky mechanism is dominant on the *hcp*-Ni@Pt$_{2L}$ surface. This prediction effectively rationalizes the experimental Tafel slope of 51 mV dec⁻¹. The DFT calculations further suggest that *hcp*-Ni@Pt$_{2L}$ is the optimal catalyst due to its superior kinetics. These predictions are in accordance with experimental observations.

It is worth noting that our calculations of the hydrogen binding energies were based on low surface coverage of H* on the Pt surfaces (coverage, 1/16). Under HER conditions, H* fully covers the Pt surface. To reveal the coverage effect on $\Delta G_{H^*}$, we carried out calculations on some benchmarks with 1 monolayer coverage of H* (Supplementary Fig. 21). The $\Delta G_{H^*}$ on $hcp$-Ni@Pt$_{2L}$ was closest to 0, indicating the highest HER performance as observed experimentally, consistent with the calculation results with low surface coverage of H*. Our DFT calculations with the same benchmarks also revealed a rough correlation between $\Delta G_{H^*}$ and the $d$-band center of the catalysts (Supplementary Fig. 22). A downshift of the $d$-band center corresponded to a decrease in $\Delta G_{H^*}$. Based on prior research and our experimental results, the closer proximity of $\Delta G_{H^*}$ to zero is indicative of improved performance in the HER. Therefore, a downshift of the $d$-band center corresponds to an increase in the HER activity, consistent with the prediction by the $d$-band center theory.

Therefore, the HER process on our $hcp$-Ni@Pt-skin catalysts can be understood as follows, based on the experiments and DFT calculations (Fig. 5e). First, the water dissociation occurs preferentially at step sites of the metastable $hcp$-Pt surface (Volmer reaction). Then, the H* migrates to the terrace $hcp$-Pt surface, where it desorbs from the Pt surface to form $H_2$ via the Heyrovsky reaction. In this reaction, the metastable $hcp$ phase of the Pt skin contributes to the lowered energy barriers in both the water dissociation and the hydrogen desorption processes. The step sites contribute to further reduced water dissociation energy barrier. The core-shell strain and electron transfer effectively modulate the electronic structure of the $hcp$-Pt skin, showing substantially decreased hydrogen binding energies that facilitate the hydrogen desorption process. As a result, the HER on the $hcp$-Ni@Pt-skin surface shows significantly boosted reaction kinetics.

## Discussion

In summary, we have demonstrated the successful synthesis of Pt skins of several atomic layers with an unconventional metastable $hcp$ phase on $hcp$-Ni nanocrystals by coherent crystal growth. The use of non-noble 3$d$ transition metal nanocrystals as the substrates significantly decreases the amount of Pt used, leading to a reduced cost. More importantly, such an approach allows for effective phase and electronic engineering of the Pt skins. We observe that the $hcp$ phase of the Ni substrates has been successfully replicated by the Pt skins on non-(0001) facets. On these facets, the Ni substrate further endows the $hcp$-Pt skin with uniaxial compressive strains and thickness-dependent electron density, leading to tailored electronic properties. Rich step sites are also observed on these $hcp$-Pt surfaces. Our experimental and DFT calculation results confirm that the $hcp$ phase, the unique electronic property modulated by the core-shell interactions, and the surface step sites effectively reduce the energy barriers associated with the water dissociation and the hydrogen desorption processes. With these features particularly preferred for the alkaline HER, the as-designed catalyst (approximately two atomic layers of $hcp$-Pt on Ni) demonstrates a high activity up to 133 mA cm$_{geo}^{-2}$ and 17.4 mA μg$_{Pt}^{-1}$ at −70 mV with 1.5 μg of Pt in 1 M KOH. Our synthesis not only opens the door to a family of core-shell nanostructured catalysts with high utilization of noble metal atoms but also provides opportunities for engineering the phase and electronic properties of ultrathin noble metal skins, both of which are highly desirable for producing efficient and cost-effective catalysts.

## Methods

### Materials

Chloroplatinic acid ($H_2PtCl_6$), nickel acetylacetonate [Ni(acac)$_2$], polyvinylpyrrolidone (PVP, Mw 55000), oleylamine, oleic acid, oleamide, formaldehyde (30% in $H_2O$), resorcinol, 1-heptanol, isopropanol, ethanol, $n$-butylamine, potassium hydroxide (KOH), hydrochloric acid

(HCl, 37 wt%), and Nafion (5 wt% in $H_2O$) were purchased from Aladdin. All chemicals were used as received without purification.

### Synthesis of $hcp$-Ni nanobranches

In a typical synthesis, 4 mL of oleylamine, 4 mL of oleic acid, 2 mL of Ni(acac)$_2$ (0.2 M in 1-heptanol), 80 μL of $H_2PtCl_6$ (0.1 M in 1-heptanol), 0.5 mL of formaldehyde (30 wt% in $H_2O$), 0.4 mL of resorcinol (1 M in 1-heptanol), and 75 mg of oleamide were added to 20 mL of 1-heptanol in a 100 mL-capacity Teflon-lined stainless autoclave and heated at 187 °C for 5 h. $hcp$-Ni nanobranches were obtained as a precipitate and redispersed in 20 mL of 1-heptanol.

### Synthesis of $hcp$-Ni@Pt-skin nanobranches

$hcp$-Ni@Pt-skin core-shell nanobranches were synthesized by a seeded growth method. Typically, 4 mL of PVP (10 wt% in 1-heptanol) and 1 mL of resorcinol (1 M in 1-heptanol) were added to 20 mL of the $hcp$-Ni nanobranches in 1-heptanol and heated to 144 °C in an N$_2$ atmosphere. A growth solution, prepared by dissolving 0.3 mL of $H_2PtCl_6$ (0.1 M) and 0.3 mL of oleylamine in 2.4 mL of 1-heptanol, was added into the reaction system with a syringe pump at a rate of 0.8 mL min$^{-1}$, i.e., 8 μmol Pt min$^{-1}$. The resulting $hcp$-Ni@Pt-skin nanobranches were collected by centrifugation and washed with ethanol. The thickness of the Pt skin depends on the amount of the growth solution: 2 mL for Ni@Pt$_{1.3}$, 3 mL for Ni@Pt$_{2.6L}$, 3.5 mL for Ni@Pt$_{3.2L}$, and 4 mL for Ni@Pt$_{4.0L}$.

### Synthesis of $fcc$-Ni@Pt-skin nanobranches and Pt nanoshells

$fcc$-Ni@Pt-skin nanobranches were obtained by heating the $hcp$-Ni@Pt-skin nanobranches in 1-heptanol at 280 °C for 20 h in an H$_2$-filled Teflon-lined autoclave. Pt nanoshells were obtained by etching the $hcp$-Ni@Pt-skin nanobranches with 1 M HCl.

### Electrochemical measurements

All measurements were taken on a CHI760e electrochemical workstation (CH Instruments) with a three-electrode configuration at 25 °C. A rotating disk electrode (RDE, 0.196 cm$^2$), a graphite electrode, and a saturated calomel electrode (SCE, calibrated before use, Supplementary Fig. 23) were used as the working, counter, and reference electrodes, respectively. The $hcp$-Ni@Pt-skin nanobranches were washed with ethanol three times, maintained statically in $n$-butylamine for 24 h, and washed with ethanol three times to remove the surface ligands[79]. The nanobranches were supported on carbon nanotubes and dispersed in a mixture of $H_2O$/isopropanol = 1:1 (volume ratio) to form a stable ink. Then, an aliquot of the ink (containing 1.5 μg of Pt for activity measurements and 3 μg of Pt for stability tests) was dropped and dried on an RDE, followed by 4 μL of Nafion (0.25 wt% in isopropanol). During the electrocatalytic measurements, the RDE was rotated at 1600 rpm to accelerate the mass transfer and effectively remove $H_2$ bubbles from the electrode surface. LSV curves were collected in N$_2$-saturated KOH (1 M, pH 13.81 ± 0.01, Supplementary Fig. 24) in the potential range of 0 to −0.10 V at a sweep rate of 10 mV s$^{-1}$ with 90% $iR$ compensation. To evaluate the durability of the catalysts, chronopotentiometry was measured at constant current densities of 10 and 100 mA cm$^{-2}$ with an L-type working glassy-carbon electrode.

### Characterizations

Low-magnification TEM images were taken on a Hitachi HT-7700 operated at 100 kV. Atomic resolution HAADF-STEM images and EELS elemental mapping analyses were performed on a JEM-ARM300F (GrandARM) equipped with double spherical aberration ($C_s$) correctors, ADF/BF/SAAF detectors, and an Electron Energy Loss Spectrometer with a direct detection camera (K2) and GIF spectrometer, operated at 300 kV. XRD patterns were recorded on a Rigaku SmartLab powder X-ray diffractometer equipped with Cu K$_\alpha$ radiation. XPS

analysis was conducted on an ESCALAB Xi+ equipped with monochromatic Al K$_\alpha$ radiation. The binding energies were calibrated using the C 1 s peak (284.6 eV). The Pt 4 $f$ peaks were fitted by Pt 4$f_{7/2}$ and 4$f_{5/2}$ doublets with a binding energy difference of 3.33 eV, an area ratio of 4/3, and the same full width at half maximum (FWHM). Quantitative elemental measurements were conducted by ICP-MS on a PerkinElmer NexION 350D.

## DFT calculations

DFT calculations were performed using the VASP (Vienna ab initio Simulation Package, version 5.4.4)[80–82]. The solvation effect has been included by VASPsol, which implements an implicit solvation model that describes the effect of electrostatics, cavitation, and dispersion on the interaction between a solute and solvent. The relative dielectric constant was set to 78.4, corresponding to water at room temperature. For reactions that can be influenced by the applied potential, such as the Heyrovsky reaction, we conducted simulations by incorporating both solvation effects and constant potential using joint density functional theory as implemented in JDFTx. The pH effect was also considered in the conversion from the standard hydrogen electrode (SHE) to the reversible hydrogen electrode (RHE) scale. The exchange-correlation interaction was described using the Perdew-Burke-Ernzerhof (PBE) functional with the Becke-Johnson damping DFT-D3 correction for London dispersion (van der Waals attraction)[83,84]. Calculations based on a benchmark demonstrate that the hydrogen binding energies by using the PBE and the B3LYP functionals were very close (Supplementary Table 4). Therefore, all the calculations in this work were carried out under the PBE level, which is ~300 times faster than the B3LYP. The projector augmented wave (PAW) method was used to account for the core-valence interactions[85]. The plane-wave energy cutoff was taken as 400 eV. The Brillouin zone was sampled based on the Monkhorst-Pack scheme with a $3 \times 3 \times 1$ $k$-point mesh. The second-order Methfessel-Paxton smearing was applied with a width of 0.1 K. eV. $E_{H^*}$ is calculated as $E_{H^*} = E_{surf+H^*} - E_{surf} - E_{H2}/2$, where $E_{surf+H^*}$ and $E_{surf}$ are the total energies of the surface with and without the hydrogen adsorbate, respectively, and $E_{H2}$ is the total energy of a hydrogen molecule. $G_{H^*}$ is calculated as $G_{H^*} = E_{H^*} + E_{ZPE} - T S_m^v$ and in this work, $T$ was set to 298.15 K. During the geometric optimizations, the two bottom layers of the atoms were fixed at their bulk positions, whereas the rest of the atoms were allowed to be relaxed in the structure.

## Data availability

The data supporting the findings of this study are available within the paper and the Supplementary Information. Source data are provided with this paper.

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

## Acknowledgements

C.G. acknowledges the support of the National Natural Science Foundation of China (22071191, 21671156), the Key Scientific and Technological Innovation Team of Shaanxi Province (2020TD-001), the Fundamental Research Funds for the Central Universities, and the Instrument Analysis Center of Xi'an Jiaotong University (for XPS and ICP-MS measurements). K.L. acknowledges the support of the Project funded by the China Postdoctoral Science Foundation (2019TQ0249) and the Natural Science Basic Research Plan in Shaanxi Province (2022JQ-100). T.C. thanks the support from Collaborative Innovation Center of Suzhou Nano Science & Technology, the Priority Academic Program Development of Jiangsu Higher Education Institutions (PAPD), the 111 Project, Joint International Research Laboratory of Carbon-Based Functional Materials and Devices, the National Natural Science Foundation of China (21903058 and 22173066), and the Natural Science Foundation of Jiangsu Higher Education Institutions (SBK20190810). Q.Z. and O.T. thank the support from the Center for High-resolution Electron Microscopy (EM-02161943) and Shanghai Key Laboratory of HREM (21DZ2260400), ShanghaiTech University, for spherical-aberration-corrected HR-STEM analysis.

## Author contributions

C.G. conceived the idea and supervised the project. K.L. conducted the synthesis, material characterizations, and electrocatalysis. H.Y. and Y.L. carried out DFT calculations. T.C. supervised the DFT calculations. Y.J., Q.Z., and O.T. conducted atomic-resolution electron microscopy characterizations and provided insightful comments. Z.L., S.Z., Z.Z., and Z.Q. assisted with the material characterizations. C.G., K.L., T.C., and Q.Z. wrote the manuscript. All authors discussed the results and revised the manuscript.

## Competing interests

The authors declare no competing interests.

## Additional information

**Peer review information** : *Nature Communications* thanks the other anonymous reviewer(s) for their contribution to the peer review of this work. Peer review reports are available.

