## [Peer review file · Nature Communications]

REVIEWER COMMENTS

Reviewer #2 (Remarks to the Author):

The revision responded to some questions from both reviewers, and improved the quality of the work. But there are still a few of major issues need to be addressed before its consideration for publication in Nature Communications.

1. The authors added HRTEM of hcp-Ni@Pt_{2.6L} after chronopotentiometric stability test (current density, 10 mA cm⁻² ; duration, 6 h) to show the material stability. 10 mA cm⁻² is a very low current density. For the real application, over 1 A cm⁻² is needed. The stability should be conducted at least 100 mA cm⁻² or larger current density.
2. When claiming “Parts of the metastable hcp-Pt skin transformed into the stable fcc phase after the long-term catalysis. However, most of the Pt-skin has retained the metastable hcp phase”, it’s better to add the statistics data. Otherwise, how to get this conclusion?
3. From exchange current density in supplementary Fig 14b, hcp-Ni@Pt and Pt show only minor difference. The authors need to comment on this. The claimed high activity was not supported by exchange current density. Also, exchange current density needs to be compared with other reported papers in a table. For the better comparison, exchange current density can use mA cm⁻² (not Pt).
4. What H or OH coverage is used in the DFT and why? The coverage effect should be added.

Reviewer #3 (Remarks to the Author):

The authors present a well-written and overall well-structured study on a novel catalyst material for the HER based on Ni nanocrystallites that have been coated with a Pt skin. The new synthesis route the authors present allows them to obtain a metastable Pt coating of two to five layers, introducing several interesting effects.

The authors outline that the resulting electronic structure of the surface, the strain induced by the underlying Ni structure and the high surface area make the new material a superior HER catalyst. They argue using data from XRD, XPS, TEM and DFT calculations as well as an electrochemical characterization of the catalyst.

Overall I would consider this an interesting and innovative approach and I think the claims by the authors are mostly comprehensible and their argumentation is sound. However, in recent years so many

studies on Pt/Ni core-shell structures have been presented (for example also for the ORR) that the material is one of many very similar materials.

Here, a detailed comparison and a discussion of the pros and cons

(for example a direct comparison to Pt/Ni core-shell nanoparticles) would be helpful and needed to underline the novelty of the catalysts.

The authors state "... to address the sluggish kinetics of many

important processes, including the ... HER under alkaline conditions for water-electrolysis ..."

I would consider the relevance of a good HER catalyst

a little overstated as the bottleneck of water splitting electrocatalysis is the OER and not the HER. While advances in HER catalysis will surely be beneficial for practical applications, there is only very little ground to be made in comparison to the optimization potential on the anode. Even for the chloralkali process, novel techniques exist like the oxygen-depolarization cathode which avoid the HER altogether.

In the discussion on the activity and stability of the catalyst

the authors discuss that it is not easy to determine an ECSA,

but as this marker is so important to compare different catalysts

on the same footing, I would at least propose to discuss this issue

in a little more detail or to attempt some other means to determine

comparable material markers (maybe in the SI).

My main criticism concerns the DFT calculations, which cannot

really be considered state of the art anymore.

If the model is supposed to describe HER in aqueous solutions, at least an implicit solvent model should be considered and in recent years the community has started to apply schemes in which the electrochemical potential is included explicitly in the model. While this would probably not be essential for the HER, which is typically carried out at low anodic potentials, it might change the energetics slightly.

Furthermore, I would expect that a high level DFT study includes results from at least two functionals including a more modern hybrid functional to assess issues like the self-interaction error of DFT and give rough error estimates for the computed values.

From the paper itself and the "methods" part it is also not clear whether the interface was fully optimized or whether just HER intermediates have been relaxed. The authors write "all the Pt and Ni atoms were fixed at their bulk positions" - this would not yield a physical description, as the structures would not represent

any local minimum that would be observable in experiment.

Furthermore, even for the HER intermediates, the approach to relax light atoms on the surface while fixing all Pt and Ni atoms is a crude approximation as, for example, the adsorption of oxygen species (H₂O, OH) might distort surface structures especially on stepped interfaces.

Here, the authors should clearly report how the interface models have been constructed and relaxed. Generally, to obtain valid models, interface structures should at least to some extent correspond to fully relaxed systems.

This also concerns the data in Fig. 4 c - how have these models been obtained? Have the interface structures been optimized, or are these just "constructed" models that only exist in silico?

In Figure 4a the authors show the barriers for water dissociation, for which they find an energy of 0.5 to 1.5 eV. This corresponds to 50-150 kJ/mol or 10-30 kcal/mol. Furthermore, the dissociation energy is reported to be in the order of 1.3 eV on fcc-Ni/Pt.

While the former needs to be compared to dissociation of water in aqueous solution, the latter can be compared to the pK_a of water. Here, a more thorough discussion of the values with results reported in the literature would be helpful to assess the results. In addition to this, it would be appropriate to study not only specific intermediates and single steps but the full HER mechanism.

Furthermore, what can be observed for the H₂O dissociation is mostly BEP behavior which would mostly be related to the different OH adsorption behavior on the interface. Here, a more detailed analysis would be helpful to clarify the underlying effects.

I would also agree with Ref 2 from the previous review (questions 15 and 16) and I would consider the authors' response that they will include "more electrochemical parameters" in the future

somewhat insufficient for publication in a high ranking journal.

This also applies to some of the data obtained for adsorption energies dissociation energy. Note that while rough trends often can be obtained in a right-for-the-wrong-reason fashion, only a more quantitative agreement in several observables can strengthen

the confidence in results obtained from theory with very approximate models.

While the authors discuss the effect of the interface structure on the electron density, the d-band center or the Fermi level position, none of this is discussed in the DFT section of the paper. Here, a thorough discussion in connection to the arguments made in the paper would help to better integrate the computed results with the reasoning in the results and discussion section of the paper.

Typos:

Page 14 line 13 "efficacy" - efficiency ?

Page 19 line 2 "We further examined, by DFT calculations, the ..." rephrase or omit comma ?

Overall, given the fact that the material the authors discuss is one Pt/Ni catalyst among many others and open questions concerning the DFT, the stability and characterization of the material, I would recommend to transfer the manuscript to a more specialized journal as it does not match the high standards of a journal like Nature Communications in its current form.

Response to Reviewers

Dear Reviewers,

Thank you very much for the valuable comments and suggestions on our manuscript, which have been very helpful for us in improving the quality of this work. We here respond point by point. All changes have been highlighted in the revised manuscript. We hope you will find that all your concerns have been appropriately addressed.

Response to Reviewer #2

The revision responded to some questions from both reviewers, and improved the quality of the work. But there are still a few of major issues need to be addressed before its consideration for publication in Nature Communications.

Response: We gratefully thank the reviewer for his/her valuable comments on our manuscript. We have supplemented new experimental data to address the remaining issues raised by the reviewer. We here respond point by point.

1. The authors added HRTEM of *hcp*-Ni@Pt_{2.6L} after chronopotentiometric stability test (current density, 10 mA cm⁻²; duration, 6 h) to show the material stability. 10 mA cm⁻² is a very low current density. For the real application, over 1 A cm⁻² is needed. The stability should be conducted at least 100 mA cm⁻² or larger current density.

Response: We thank the reviewer for the suggestion. We agree that real water-splitting hydrogen production is conducted at a much higher current density than 10 mA cm⁻². Following the reviewer's suggestion, we have conducted the chronopotentiometric measurement of the *hcp*-Ni@Pt-skin catalyst at 100 mA cm⁻² and examined the structure of the catalyst after the catalysis. The results have been added to **Fig. 3f** of the revised manuscript and the revised Supplementary **Fig. 18**.

The chronopotentiometric measurement of the *hcp*-Ni@Pt_{2.6L} nanobranches showed satisfactory stability at the current density of 100 mA cm⁻². The overpotential gradually increased from 80 to 230 mV in 50 h. The nanobranches after catalysis (100 mA cm⁻², 20 h) were subjected to careful *C_s*-corrected HAADF-STEM imaging. We collected 10 high-resolution images from different domains of the nanobranches. These images showed that the *hcp* phase, the {01-11} facets, and the step sites of the Pt skins have been retained. The nanobranches showed a relatively rough surface with less-uniform thicknesses of the Pt skins, suggesting a tendency of Pt atom migration on the Ni surface to form larger assemblies during the long-term catalysis. Therefore, our overall conclusion is that the *hcp*-Ni@Pt nanobranches are stable, which accounts for the catalytic durability. Some structural changes can be occasionally observed, which contributed to the overpotential increase during the long-term stability.

Fig. 3 (f): Chronopotentiometric curves of the *hcp*-Ni@Pt_{2.6L} at a constant current density of 10 and 100 mA cm_{geo}⁻². The chronopotentiometric curve of the Pt/C was also listed for comparison. Inset: Chronopotentiometric curves in the first 10 h.

Supplementary Fig. 18 | Structural analysis of the *hcp*-Ni@Pt_{2.6L} core-shell nanobranched after the chronopotentiometric stability test at 100 mA cm⁻² for 20 h. Two nanobranched were examined domain by domain. (a, g) HAADF-STEM images of the respective nanobranched. (b–f, h–l) C_s -corrected

high-resolution HAADF-STEM images. The Pt-skins on the sides of the nanobranched have retained the metastable *hcp* phase. The {1-101} facets and step sites were also retained. At the tip of the nanobranched, Pt-skin retained its *fcc* phase (k). It can be concluded that the *hcp*-Ni@Pt nanobranched showed high structural stability during the electrocatalytic HER. It may be because the phase change involves the position change of a large number of atoms, which is a difficult process with a high energy barrier. Therefore, the phase change can only be observed occasionally in small local areas as shown in Supplementary Fig. 17. By statistically analyzing all the high-resolution images available, including the 10 images in Supplementary Fig. 18 and the one image in Supplementary Fig. 17, the fraction of the phase change during the catalysis can be roughly estimated to be ~3% (*hcp*, 635 atomic lines; *fcc*, 21 atomic lines).

The following discussion has been added to the revised manuscript.

Page 17: “We also investigated the catalytic stability of the *hcp*-Ni@Pt_{2.6L} nanobranched at a high current density of 100 mA cm_{geo}⁻². The overpotential showed a slow increase from 80 to 230 mV in 50 h, suggesting satisfactory catalytic stability.” “The fine structure of the *hcp*-Ni@Pt_{2.6L} nanobranched after catalysis was further inspected by high-resolution TEM and C_s-corrected HAADF-STEM (Supplementary Figs. 17 and 18). By carefully examining the crystal structures of the nanobranched at different domains, we can statistically conclude that the metastable *hcp* phase has been largely retained during the long-term electrocatalysis even at the high current density. We also observed minor local areas that transformed into the more stable *fcc* phase, which, however, can only be observed occasionally. Statistically, only ~3% of the *hcp* phase was transformed into the more-stable *fcc* phase on the side facets of the nanobranched. Moreover, the high-resolution electron microscopy images also confirmed that the {01-11} facets and the step sites of the Pt skins were largely retained. The structural integrity of the *hcp*-Ni@Pt core-shell nanobranched contributes to the catalytic durability. Compared with the pristine *hcp*-Ni@Pt_{2.6L} nanobranched, the ones after the catalysis showed a relatively rough surface with less-uniform thicknesses of the Pt skins, suggesting a tendency of the Pt atoms to migrate on the Ni surface to form large ensembles (Supplementary Fig. 18). The surface reconstruction may have altered the electronic structure of the Pt skin to an extent, which contributes to the observed increase in the overpotential during the long-term catalysis. Moreover, a large number of H₂ bubbles were observed to form on the electrode surface due to the high current density (Supplementary Fig. 19). The bubbles partially covered the catalyst surface and possibly peeled off the catalyst from the electrode surface, which may also have contributed to the overpotential increase in the catalysis.”

2. When claiming “Parts of the metastable *hcp*-Pt skin transformed into the stable *fcc* phase after the long-term catalysis. However, most of the Pt-skin has retained the metastable *hcp* phase”, it’s better to add the statistics data. Otherwise, how to get this conclusion?

Response: We thank the reviewer for the valuable comments. We are sorry for not providing sufficient data to support this claim in our previous revision. We agree that the phase of the Pt skin should be

analyzed statistically so that the extent of the phase retention can be described in a quantitative way.

During the revision, we carefully analyzed the *hcp*-Ni@Pt-skin after the chronopotentiometric durability test at 100 mA cm⁻² for 20 h. Two nanobranches were examined domain by domain. Ten atomic-resolution C_s-corrected HAADF-STEM images were collected. These images are presented in **Fig. 18** of the revised Supplementary Information. By carefully examining the atomic arrangements in these 10 images, we concluded that all the Pt-skins on the sides of the nanobranches retained the metastable *hcp* phase. This may be because the phase change involves the position change of a large number of atoms, which is a difficult process with a high energy barrier. Therefore, the phase change can only be observed occasionally in small local areas, as shown in Supplementary **Fig. 17**. By statistically analyzing all the high-resolution images available, including the 10 images in Supplementary **Fig. 18** and the one image in Supplementary **Fig. 17**, the fraction of the phase change can be roughly estimated to be ~3% (*hcp*, 635 atomic lines; *fcc*, 21 atomic lines).

The corresponding discussion on the statistical analysis has been added to the caption of Supplementary **Fig. 18** and the revised manuscript as copied below:

Page 17: “Statistically, only ~3% of the *hcp* phase was transformed into the more-stable *fcc* phase on the side facets of the nanobranches.”

3. From exchange current density in supplementary Fig 14b, *hcp*-Ni@Pt and Pt show only minor difference. The authors need to comment on this. The claimed high activity was not supported by exchange current density. Also, exchange current density needs to be compared with other reported papers in a table. For the better comparison, exchange current density can use mA cm⁻² (not Pt).

Response: We thank the reviewer for the valuable comments. We also appreciate the reviewer for the kind suggestion of using mA cm⁻² (not Pt) for calculating the exchange current densities.

The exchange current densities (j_0) in our previous submission were derived from the HER polarization curves in the potential range of -59 ~ -80 mV. In this potential range, the j_0 of the *hcp*-Ni@Pt showed only a two-fold increase compared with the Pt/C. However, the Tafel slopes in this potential range differ significantly (*hcp*-Ni@Pt, 47~65 mV dec⁻¹; Pt/C, 131 mV dec⁻¹). The Tafel slope is a measure of the potential increase for achieving a 10-fold increase in the current density. This means that although the j_0 (oxidation/reduction current density at an overpotential of 0) are comparable, the current density (j) with the *hcp*-Ni@Pt catalyst can be increased at a rate that is 10²~10³ times greater than the Pt/C in response to an applied overpotential. Therefore, the kinetics of the HER has been significantly boosted with the *hcp*-Ni@Pt catalyst at an applied potential.

To better investigate the exchange current densities of the catalysts, we here adopt an alternative method for the calculation (A. J. Bard, L. R. Faulkner, Electrochemical Methods: Fundamentals and

Applications, 2nd Edition). We consider an HER process containing two electron-transfer steps (number of transferred electrons, $n=2$). The Butler-Volmer equation can be described as $j = j_0[e^{-(n'+\alpha)f\eta} - e^{(n''+1-\alpha)f\eta}]$, where j is the current density, j_0 is the exchange current density, α is the transfer coefficient, $f = F/RT$ is a constant, η is the overpotential, n' and n'' are the numbers of the transferred electrons before and after the rate-determining step, $n'+n''+1 = n$. This equation can be changed to $j = j_0 e^{-(n'+\alpha)f\eta} (1 - e^{nf\eta})$ and further to $\log_{10}[j/(1 - e^{nf\eta})] = \log_{10} j_0 - [(n'+\alpha)F/(2.303RT)] \eta$. Therefore, the exchange current density (j_0) can be derived by plotting $\log_{10}[j/(1 - e^{nf\eta})]$ against η . This allows us to investigate the reaction kinetics in all overpotential ranges. The results have been presented in revised Supplementary Fig. 14. Clearly, the $\log_{10}[j/(1 - e^{nf\eta})] \sim \eta$ plots of the *hcp*-Ni@Pt catalysts showed a two-phase feature: When the overpotential is lower than -75 mV, the current densities were low but the slopes were high; when the overpotential is higher than -75 mV, the current densities were high but slopes were low. This indicates different HER mechanisms in these two overpotential ranges.

Supplementary Fig. 14 | Calculation of the exchange current densities with different catalysts. The current densities were normalized to the geometric area of the electrodes.

For better comparison with the reported values in the literature, we calculated the current densities at relatively large overpotentials (potential range, $-75 \sim -95$ mV) (Supplementary Fig. 14). All the *hcp*-Ni@Pt catalysts showed substantially higher exchange current densities than the Pt/C. In particular, the exchange current density of the *hcp*-Ni@Pt_{2.6}L catalyst was calculated to be $3.44 \text{ mA cm}_{\text{Pt}}^{-2}$, which is ~ 8 times greater than that of the commercial Pt/C, confirming the boosted kinetics of the HER.

We also listed some of the exchange current densities reported in the literature (Supplementary Table 1, only values obtained under similar standards are listed). It can be inferred that our values are higher than the reported values, confirming the superiority of the *hcp*-Ni@Pt catalyst in achieving exceptional catalytic activities in the alkaline HER.

4. What H or OH coverage is used in the DFT and why? The coverage effect should be added.

Response: We thank the reviewer and agree that the surface coverages of H^* and *OH are important and can significantly influence the HER reaction. Previous research suggests that underpotential deposition (UPD) of hydrogen readily occurs on the Pt surface and leads to the saturation of the H^* surface coverage under the HER conditions (*Nat. Catal.* 2022, 5, 923). Our previous calculations were based on low H^* surface coverage ($\theta = 1/16$). To further study the effect of the H^* coverage, we additionally performed simulations at 1 ML coverage of H^* . The new results have been added to the revised Supplementary Fig. 21. The results indicate that the hydrogen binding energy (HBE) of *hcp*-Ni@Pt_{2L} is closest to zero, which is favorable for HER. This conclusion is in agreement with our earlier calculations at low H^* surface coverage ($\theta = 1/16$).

Supplementary Fig. S21 | Binding energies of hydrogen on *hcp*-Ni@Pt_{2L}, *hcp*-Ni@Pt_{2L} with step sites, and *fcc*-Ni@Pt_{2L} surfaces at low (a, H^* coverage = 1/16) and high (b, H^* coverage = 1) coverages of H^* . The DFT results suggest that the coverage of H^* on the catalyst surface can affect the binding energy of hydrogen. However, the general trend is similar. In both calculations, the binding energy of hydrogen on *hcp*-Ni@Pt_{2L} was closest to 0, indicating the best HER performance, consistent with experimental observations.

Page 22: “It is worth noting that our calculations of the hydrogen binding energies were based on the low surface coverage of H^* on the Pt surfaces (coverage, 1/16). Under HER conditions, H^* fully covers the Pt surface. To reveal the coverage effect on the ΔG_{H^*} , we carried out calculations on some benchmarks with 1 monolayer coverage of H^* (Supplementary Fig. 21). The ΔG_{H^*} on *hcp*-Ni@Pt_{2L} was closest to 0, indicating the best HER performance as observed experimentally, consistent with the calculation results with the low surface coverage of H^* .”

Response to Reviewer #3

The authors present a well-written and overall well-structured study on a novel catalyst material for the HER based on Ni nanocrystallites that have been coated with a Pt skin. The new synthesis route the authors present allows them to obtain a metastable Pt coating of two to five layers, introducing several

interesting effects. The authors outline that the resulting electronic structure of the surface, the strain induced by the underlying Ni structure and the high surface area make the new material a superior HER catalyst. They argue using data from XRD, XPS, TEM and DFT calculations as well as an electrochemical characterization of the catalyst. Overall I would consider this an interesting and innovative approach and I think the claims by the authors are mostly comprehensible and their argumentation is sound.

Response: We gratefully thank the reviewer for his/her recognition of our synthesis route, the structure, the interesting effects introduced by the unique structure, and the manuscript writing. We are also grateful to the reviewer for his/her valuable comments. We here respond point by point.

However, in recent years so many studies on Pt/Ni core-shell structures have been presented (for example also for the ORR) that the material is one of many very similar materials. Here, a detailed comparison and a discussion of the pros and cons (for example a direct comparison to Pt/Ni core-shell nanoparticles) would be helpful and needed to underline the novelty of the catalysts.

Response: We thank the reviewer for reminding us of the previous studies on Pt/Ni core-shell structures and offering the suggestion to highlight the novelty of our material through additional discussion.

We agree that there have been a lot of bimetallic Pt-Ni catalysts, especially for the ORR, but most of the catalysts were Pt-Ni alloy catalysts. We did thorough literature research and found that there are some reports on Ni@Pt core-shell nanostructures, which were usually synthesized by sequential chemical reduction of Ni and Pt (ref. 40–47). This synthesis involves the galvanic replacement reaction between the Ni nanocrystals and the Pt salt due to the intrinsic redox potential difference, which leads to a destruction of a clear Ni/Pt boundary, significant Ni-Pt mixing, and a loss of control over the structure. There have not been full characterizations of the nominal core-shell nanostructure to show well-defined Ni cores and Pt shells as such with defined atomic thicknesses of the shells and surface compositions.

Our materials are different from the previous nominal Ni@Pt core-shell materials. For the first time, our synthesis fully prevents the galvanic replacement between the Ni nanocrystals and the Pt salt, leading to Ni@Pt core-shell nanostructures with clear Ni/Pt boundaries and precisely controlled Pt shell thickness ranging from 2 to 5 atomic layers. In fact, our group has endeavored many efforts and proposed a seminal reduction-potential-engineering strategy for preventing the galvanic replacement reaction between Ag nanocrystals and noble metal salts (ref. 48–52). This work significantly extends this strategy to the galvanic-replacement-free growth of a noble metal (i.e., Pt) on non-noble 3d-transition metal (i.e., Ni) nanocrystals, leading to a well-defined Ni@Pt core-shell nanostructure fundamentally different from the previous ones.

Thanks to the Ni@Pt core-shell nanostructure with atomic precision, we are able to introduce many interesting effects. (1) Pt has been obtained as a novel metastable *hcp* phase by the galvanic-

replacement-free epitaxial growth of Pt on the *hcp*-Ni cores. The metastable phase has proven efficient in promoting H₂O dissociation and H₂ evolution in an alkaline HER, leading to excellent activities. (2) The electronic properties of the *hcp*-Pt have been effectively modulated by the Ni core through the progressive core-shell electron transfer and the compressive strains caused by the core-shell lattice mismatch, both being favorable for improving the HER performance.

Following the reviewer's suggestion, we have added the following discussion to highlight the novelty of this work.

Page 6: "A careful literature search suggests that Ni@Pt core-shell nanostructures have been reported previously.⁴⁰⁻⁴⁷ However, these nanostructures were usually synthesized by sequential reduction of Ni and Pt without delicate design for suppressing the galvanic replacement side-reaction, which may lead to a destruction of the Ni/Pt boundary, the significant Ni-Pt atomic mixing, and a loss of control over the structure. As a result, many of these prior works failed to provide unambiguous evidence to prove a well-defined core-shell structure as a heterojunction of monometallic Ni and Pt with sufficient control."

The authors state "... to address the sluggish kinetics of many important processes, including the ... HER under alkaline conditions for water-electrolysis ..." I would consider the relevance of a good HER catalyst a little overstated as the bottleneck of water splitting electrocatalysis is the OER and not the HER. While advances in HER catalysis will surely be beneficial for practical applications, there is only very little ground to be made in comparison to the optimization potential on the anode. Even for the chloralkali process, novel techniques exist like the oxygen-depolarization cathode which avoid the HER all together.

Response: Thanks for the valuable comments. We agree that the OER at the anode is the bottleneck of water splitting, which needs great efforts to develop effective oxygen evolution catalysts. We also believe highly efficient HER catalysts are still required for minimizing the use of Pt in practical applications, which has attracted significant attention in recent years. We apologize for overstating the significance of the HER to the overall water-splitting hydrogen production in our previous submission. We have reworded this discussion (copied below) and hope for the reviewer's approval.

Page 3: "The electrocatalytic hydrogen evolution reaction (HER) has attracted great attention because it is associated with the water-splitting production of H₂, a clean alternative to traditional fossil fuels.¹⁻⁵"

In the discussion on the activity and stability of the catalyst the authors discuss that it is not easy to determine an ECSA, but as this marker is so important to compare different catalysts on the same footing, I would at least propose to discuss this issue in a little more detail or to attempt some other means to determine comparable material markers (maybe in the SI).

Response: We thank the referee for the valuable comments. We admit that the ECSA of our *hcp*-Ni@Pt

cannot be easily obtained. This is because ECSAs are usually estimated by hydrogen underpotential deposition (UPD), Cu UPD, and CO stripping. All these techniques require that the cyclic voltammetry be swept to positive potentials. Under positive potentials, however, our *hcp*-Ni@Pt catalyst undergoes a rapid surface reconstruction, leading to the migration of the Ni atoms to the surface and the subsequent formation of nickel oxide. Moreover, gas physisorption can also provide surface area information. However, the calculation is imprecise for electrocatalysts and thus not usually used in practice.

We agree that the ECSA is a common marker to estimate the activity of electrocatalysts. Although we cannot obtain the ECSA values precisely by experiments, we propose a method to get an approximate value by involving geometric calculations. Because we know the exposing facet of the *hcp*-Pt, i.e., {01-11}, we can calculate the atom density of Pt on these surfaces, i.e., each atom occupies $7.21 \times 10^{-20} \text{ m}^2$. We also know the atomic layer number (n) of Pt by C_s -corrected HAADF-STEM imaging. Then, we can derive that $\text{ECSA} = 222.8/n \text{ (m}^2 \text{ g}^{-1}\text{)}$. By this means, the ECSAs of the *hcp*-Ni@Pt $_{nL}$ ($n = 1.3, 2.6, 3.2,$ and 4.0) are 171, 85.7, 69.6, and 55.7 $\text{m}^2 \text{ g}^{-1}$, respectively. The results have been included in the discussion on **pages 16–18** and **Table 2** of the revised Supplementary Information.

Supplementary Table 2 | ECSAs and specific activities of the *hcp*-Ni@Pt $_{nL}$ nanobranched in the alkaline HER, compared with those of Pt/C.

Sample	ECSA ($\text{m}^2 \text{ g}^{-1}$)	Specific activity at -70 mV (mA cm^{-2})
hcp -Ni@Pt $_{1.3L}$	171	1.35
hcp -Ni@Pt $_{2.6L}$	85.7	26.4
hcp -Ni@Pt $_{3.2L}$	69.6	21.3
hcp -Ni@Pt $_{4.0L}$	55.7	15.9
Pt/C	54.6	2.0

Following the reviewer’s suggestion, we have made additional discussion on the ECSA and specific activities of the *hcp*-Ni@Pt catalysts in the revised manuscript, as copied below:

Page 15: “It is difficult to measure the electrocatalytically active surface areas (ECSAs) of the catalysts directly by the underpotential deposition (UPD) of hydrogen because the Ni cores are prone to oxidation at positive potentials. However, given the atomic layer number (n) of Pt obtainable statistically by C_s -corrected HAADF-STEM imaging, the ECSA of the *hcp*-Ni@Pt $_{nL}$ {01-11} catalysts can be estimated to be $222.8/n \text{ m}^2 \text{ g}^{-1}$ by geometrical calculations (Supplementary Fig. 13). By this means, the ECSAs of the *hcp*-Ni@Pt $_{nL}$ ($n = 1.3, 2.6, 3.2,$ and 4.0) catalysts were calculated to be 171, 85.7, 69.6, and 55.7 $\text{m}^2 \text{ g}^{-1}$, respectively. With these values, the specific activities of the catalysts at -70 mV were estimated to be 1.35, 26.4, 21.3, and 15.9 $\text{mA cm}_{\text{Pt}}^{-2}$, respectively (Supplementary Table 2). Alternatively, the specific activity can be evaluated by multiplying the mass activity by the atomic layer number of the Pt skin, which contains fewer assumptions in the calculation (details, Supplementary Information) (Figure 3d). A similar volcanic trend of the specific activity can be observed with decreasing thickness of the Pt skin. The catalyst with a Pt skin of 2.6 atomic layers shows the highest activity in the alkaline HER.”

My main criticism concerns the DFT calculations, which cannot really be considered state of the art anymore. If the model is supposed to describe HER in aqueous solutions, at least an implicit solvent model should be considered and in recent years the community has started to apply schemes in which the electrochemical potential is included explicitly in the model. While this would probably not be essential for the HER, which is typically carried out at low anodic potentials, it might change the energetics slightly.

Response: We are grateful to the reviewer for offering the insightful comments. We concur that the impacts of solvation and applied potential on the DFT calculations are indeed significant. Upon review, we have identified the oversight in our previous response, which may have caused confusion for the reviewer. To clarify, we did incorporate the implicit solvent model in our DFT calculations. In response to the reviewer's constructive suggestions, we have now included the necessary information in the revised Methods section.

Page 25: "The solvation effect has been included by VASPsol, which implements an implicit solvation model that describes the effect of electrostatics, cavitation, and dispersion on the interaction between a solute and solvent. The relative dielectric constant was set to 78.4, corresponding to water at room temperature."

As commented by the reviewer, the applied potential may cause a slight change in the energetics of the HER. Following the referee's suggestion, we have applied an overpotential of -70 mV to primary reaction steps of the HER that can be influenced by the applied potential, such as the Heyrovsky reaction. The updated results have been added to revised Fig. 4b. We also included the following calculation details in the revised Methods.

Page 25: "For reactions that can be influenced by the applied voltage, such as the Heyrovsky reaction, we conducted simulations by incorporating both solvation effects and constant potential using joint density functional theory as implemented in JDFTx. The pH effect was also considered in the conversion from the standard hydrogen electrode (SHE) to the reversible hydrogen electrode (RHE) scale."

Furthermore, I would expect that a high level DFT study includes results from at least two functionals including a more modern hybrid functional to assess issues like the self-interaction error of DFT and give rough error estimates for the computed values.

Response: We agree with the reviewer and understand the reviewer's concern regarding the accuracy level of the DFT calculations. Although PBE, which we use for this study, is the most widely used functional in electrochemical calculations, its accuracy is not naturally guaranteed. Therefore, we benchmarked its performance with two hybrid functionals, B3LYP and PBE0, specifically on hydrogen

binding energy (HBE). According to the results presented in the revised Supplementary Table 4, the HBE predicted by the PBE is comparable to that predicted by the B3LYP (both approaches predict an HBE that is ~ 0.25 eV lower than that predicted with the PBE0 function). Based on these benchmark data and the consistency of the PBE prediction with the experimental results, we conclude that the calculation at the PBE level provides the optimal compromise between the efficiency and the accuracy.

Supplementary Table 4 | Comparison of the hydrogen binding energies on a benchmark calculated with different DFT functionals.

Functional	Hydrogen Binding Energy / eV
PBE	-0.50
B3LYP	-0.53
PBE0	-0.76

We have added the following discussion to the Methods section of the revised manuscript:

Page 25: “Calculations based on a benchmark demonstrate that the hydrogen binding energies by using the PBE and the B3LYP functionals were very close (Supplementary Table 4). Therefore, all the calculations in this work were carried out under the PBE level, which is ~ 300 times faster than the B3LYP.”

From the paper itself and the "methods" part is also not clear whether the interface was fully optimized or whether just HER intermediates have been relaxed. The authors write "all the Pt and Ni atoms were fixed at their bulk positions" - this would not yield a physical description, as the structures would not represent any local minimum that would be observable in experiment. Furthermore, even for the HER intermediates, the approach to relax light atoms on the surface while fixing all Pt and Ni atoms is a crude approximation as, for example, the adsorption of oxygen species (H_2O , OH) might distort surfaces structures especially on stepped interfaces. Here, the authors should clearly report how the interface models have been constructed and relaxed. Generally, to obtain valid models, interface structures should at least to some extent correspond to fully relaxed systems. This also concerns the data in Fig. 4 c - how have these models been obtained? Have the interface structures been optimized, or are these just "constructed" models that only exist in silico?

Response: We agree with the reviewer that the interface structure should be relaxed for the DFT calculations. We indeed followed this principle in our calculations. We have 4 layers of metal atoms in our model. The bottom two layers were fixed to emulate the bulk lattice in our real nanocrystals. The surface two layers of the metal atoms and the hydrogen atoms were fully relaxed during our simulations. We apologize for any confusion caused by the misleading description in our manuscript.

We have added the following description of the model in the revised manuscript:

Page 18: “The bottom two layers of the metal atoms were fixed according to the lattice parameters of *hcp*-Ni measured experimentally by electron microscopy ($a, b = 2.65 \text{ \AA}$, $c = 4.32 \text{ \AA}$), while all the other atoms were relaxed during the energy minimization. The vacuum layers were set up at least 15 \AA to minimize possible interactions between the replicated cells. The atomic coordinates with constraints are listed in Supplementary Information in the VASP format.”

Page 26, Methods Section: “During the geometric optimizations, the two bottom layers of the atoms were fixed at their bulk positions, whereas the rest of the atoms were allowed to be relaxed in the structure.”

We would like to provide further clarification regarding the models depicted in **Fig. 4c**. The models were similarly constructed, except that the bottom two layers of the metal atoms were fixed at different lattice sizes, deviating by 1%, 3%, and 5% from the intrinsic size of the *hcp*-Pt. It is worth noting that the intrinsic lattice size of *hcp*-Pt was derived from *fcc*-Pt, assuming both are close-packed structures. The model with 5% compressive strain best represents the case investigated experimentally in this work.

Following the reviewer’s suggestion, the following statement has been added to the revised manuscript:

Page 21: “During the simulations, the positions of the bottom two layers of the metal atoms were fixed with various lattice sizes. The specific lattice sizes correspond to different compressive strains (1%, 3%, and 5%) relative to the intrinsic lattice size of Pt. All the other atoms were allowed to undergo relaxation during the energy minimization.”

In Figure 4a the authors show the barriers for water dissociation, for which they find an energy of 0.5 to 1.5 eV. This corresponds to 50-150 kJ/mol or 10-30 kcal/mol. Furthermore, the dissociation energy is reported to be in the order of 1.3 eV on *fcc*-Ni/Pt. While the former needs to be compared to dissociation of water in aqueous solution, the latter can be compared to the pK_a of water. Here, a more thorough discussion of the values with results reported in the literature would be helpful to assess the results. In addition to this, it would be appropriate to study not only specific intermediates and single steps but the full HER mechanism. Furthermore, what can be observed for the H_2O dissociation is mostly BEP behavior which would mostly be related to the different OH adsorption behavior on the interface. Here, a more detailed analysis would be helpful to clarify the underlying effects.

Response: We agree with the reviewer and have added the discussions by comparing our energy barriers to the reported values. From our calculation, the energy barriers for the water dissociation (Volmer reaction) and hydrogen desorption (Tafel reaction) are in the range of 0.63-1.41 eV and 0.78-1.0 eV, respectively. These values are close to the energy barriers reported in the literature as listed in the revised Supplementary Table 3.

Supplementary Table 3 | Comparison of the reaction barriers in our results with those reported in

the literature.

Systems	Energy Barrier / eV	Reaction Types	Reference
hcp -Ni@Pt _{2L} -steps	0.63/0.83	Volmer/Heyrovsky	This Work
PtNi alloy	0.50–0.80	Volmer	6
Ru(111)	0.68/0.96	Volmer/Heyrovsky	16
Rh(111)	0.95/1.12	Volmer/Heyrovsky	16
Pt(111)	0.80/1.02	Volmer/Heyrovsky	16
RhRu alloy	0.45/0.75	Volmer/Heyrovsky	16
TMs-Ti ₂ C	0.5–1.5	Heyrovsky	17

The following discussion has been added to the revised manuscript:

Page 19: “All the energy barriers obtained from our calculations are close to the reported values (Supplementary Table 3).^{65,77,78}”

Furthermore, the reviewer proposed a comprehensive examination of the displayed relationship regarding the BEP. Indeed, as shown in Figure 4a, the reaction energy is directly proportional to the reaction site’s energy barrier. Furthermore, our comparison of the adsorption energy of OH demonstrated a positive correlation between the *OH adsorption energy and the reaction energy, as well as the reaction potential barrier. These findings indicate that an increase in the adsorption energy of OH can enhance the Volmer reaction in alkaline environments.

Following the reviewer’s suggestion, we have conducted a comprehensive investigation of the full reaction pathway of the HER to gain deeper insight into the underlying effects. The results are presented in **Figs. 4a, 4b** and Supplementary **Fig. 20**.

Fig. 4 (a, b) Free energy diagrams of water dissociation (Volmer reaction) and hydrogen desorption (Heyrovsky reaction) at -70 mV on *fcc*-Ni@Pt_{2L}, *hcp*-Ni@Pt_{2L}, and *hcp*-Ni@Pt_{2L} with step sites, respectively. Inset: Model structures after DFT optimizations.

Supplementary Fig. 20 | Free energy diagrams of the Tafel reaction (a) and the Heyrovsky reaction (b) at -70 mV on *hcp*-Ni@Pt_{2L}, *hcp*-Ni@Pt_{2L} with step sites, and *fcc*-Ni@Pt_{2L} surfaces. On the *fcc*-Ni@Pt_{2L} surface, the energy barriers for the Tafel reaction and the Heyrovsky reaction were almost identical. Considering that the Heyrovsky reaction involves the solvent of H₂O in large excess as a reactant, it may become a more favorable process for hydrogen desorption, which is consistent with the experimental Tafel slope of 47 mV dec⁻¹. The results in (b) also indicate that the metastable *hcp* phase contributes to the decreased energy barrier of the Heyrovsky reaction, compared with the *fcc* counterpart.

In brief, **Fig. 4a** is the free energy diagram of water dissociation (Volmer reaction). The metastable *hcp* phase and the step sites on its surface were found effective in reducing its energy barrier. The subsequent H₂ formation prefers the Heyrovsky reaction compared with the Tafel reaction (Supplementary **Fig. 20**). On the *fcc*-Ni@Pt_{2L} surface, the energy barriers for the Tafel reaction and the Heyrovsky reaction were almost identical. Considering that the Heyrovsky reaction involves the solvent of H₂O in large excess as a reactant, it may become a more favorable process for hydrogen desorption. Such a Volmer-Heyrovsky mechanism is consistent with the experimental Tafel slope of 47 mV dec⁻¹. By analyzing the energy barriers related to the Volmer and Heyrovsky reactions, we determined that the rate-limiting step underwent a transition from the Volmer reaction to the Heyrovsky reaction when the phase of Pt changed from the *fcc* to the metastable *hcp*. Please see below the extensive discussion on the full-path DFT results in the revised manuscript for details.

Page 18: “Fig. 4a shows the free energy diagram of water dissociation (Volmer reaction) on *hcp*-Ni@Pt_{2L}, *hcp*-Ni@Pt_{2L} with step sites, and *fcc*-Ni@Pt_{2L}. The Volmer reaction on *fcc*-Ni@Pt_{2L} needs to overcome the highest energy barrier (1.41 eV). In contrast, the *hcp*-Ni@Pt_{2L} showed a significantly reduced energy barrier (0.84 eV), suggesting that the metastable *hcp* phase played a vital role in reducing the energy barrier of the Volmer reaction. The step sites on the *hcp*-Ni@Pt{01-10} surface further decreased this energy barrier to 0.63 eV, confirming that the step sites are particularly active sites for the Volmer reaction. We also investigated the free energy diagrams of the H₂ formation, which can proceed via either the Heyrovsky-type reaction or the Tafel-type reaction. DFT calculation suggests that the Heyrovsky reaction is more favorable (Supplementary Fig. 20). As shown in Fig. 4b, the Heyrovsky reaction on the *hcp*-Ni@Pt_{2L} surface shows an energy barrier of 0.78 eV at -70 mV, which was lower than the energy barrier on the *fcc*-Ni@Pt_{2L} surface (1.0 eV) by 0.22 eV, suggesting that the metastable

hcp phase also substantially reduced the energy barrier of the Heyrovsky reaction. All the energy barriers obtained from our calculations are close to the reported values in the literature.^{65,77,78} By comparing the energy barriers in the full HER pathway (Fig. 4a, b), we can infer that the rate-determining step of the HER was the Volmer reaction on the *fcc*-Ni@Pt_{2L} surface and shifted to the Heyrovsky reaction on the *hcp*-Ni@Pt_{2L} surface, which well agrees with our experimental Tafel analysis. With the latter becoming the rate-determining step, the HER kinetics is substantially accelerated due to the reduced overall energy barrier. It is worth noting that although the step sites on the *hcp*-Ni@Pt_{2L} surface are particular hotspots for the Volmer reaction, the subsequent Heyrovsky reaction at these sites suffers from a higher energy barrier, suggesting that a hydrogen diffusion to the adjacent terrace surface is necessary prior to H₂ formation via the Heyrovsky reaction.”

I would also agree with Ref 2 from the previous review (questions 15 and 16) and I would consider the authors' response that they will include “more electrochemical parameters” in the future somewhat insufficient for publication in a high ranking journal. This also applies to some of the data obtained for adsorption energies dissociation energy. Note that while rough trends often can be obtained in a right-for-the-wrong-reason fashion, only a more quantitative agreement in several observables can strengthen the confidence in results obtained from theory with very approximate models.

Response: We thank the reviewer for the suggestions. In our simulation, we systematically considered the solvation effect using the implicit solvation model as implemented in VASPsol. For reactions (such as the Heyrovsky reaction) that are sensitive to the applied potential, we considered both the solvation effect and constant potential simulation using joint density functional theory as implemented in JDFTx. The pH effect is also included in the transformation from the standard hydrogen electrode (SHE) to the reversible hydrogen electrode (RHE) scale. For the reactions, we considered both thermodynamics and kinetics.

We have added the following discussion about the simulation methods in the revised manuscript:

Page 25: “The solvation effect has been included by VASPsol, which implements an implicit solvation model that describes the effect of electrostatics, cavitation, and dispersion on the interaction between a solute and solvent. The relative dielectric constant was set to 78.4, corresponding to water at room temperature. For reactions that can be influenced by the applied potential, such as the Heyrovsky reaction, we conducted simulations by incorporating both solvation effects and constant potential using joint density functional theory as implemented in JDFTx. The pH effect was also considered in the conversion from the standard hydrogen electrode (SHE) to the reversible hydrogen electrode (RHE) scale.”

These simulation techniques, i.e., the efficient and accurate methods, are utilized to simulate electrochemical reactions. By utilizing these precise methods, comparisons with experimental results can be made with greater accuracy, leading to more consistent predictions. A quick summary is as

follows:

- (1) The DFT-predicted reaction mechanisms are consistent with the experiment. After exploring all the possible full HER reaction mechanisms, the DFT calculations suggest a Volmer-Heyrovsky mechanism on the *hcp*-Ni@Pt_{2L} surface. Such a prediction can well explain the experimentally observed Tafel slope of 47 mV dec⁻¹.
- (2) The DFT calculations suggest *hcp*-Ni@Pt_{2L} as the best catalyst because of the preferred kinetics. Such a prediction well agrees with the experiment.
- (3) The hydrogen binding energies at elevated surface coverage (1 ML) are qualitatively consistent with the trend of the experimental performances.

In addition, following the reviewer's suggestion, we also include the *d*-band center analysis, which can well explain the performances observed experimentally.

We have included the following discussions comparing the simulation and experiment:

Page 21: "Upon a comprehensive exploration of all potential full HER reaction mechanisms, DFT calculations indicate that the Volmer-Heyrovsky mechanism is dominant on the *hcp*-Ni@Pt_{2L} surface. This prediction effectively rationalizes the experimental Tafel slope of 47 mV dec⁻¹. The DFT calculations further suggest that *hcp*-Ni@Pt_{2L} is the optimal catalyst due to its superior kinetics. These predictions are in accordance with experimental observations."

While the authors discuss the effect of the interface structure on the electron density, the *d*-band center or the Fermi level position, none of this is discussed in the DFT section of the paper. Here, a thorough discussion in connection to the arguments made in the paper would help to better integrate the computed results with the reasoning in the results and discussion section of the paper.

Response: We appreciate the valuable suggestions offered by the reviewer, and agree that the *d*-band center is a useful descriptor for the metal-adsorbate interactions. Our calculations revealed that the *d*-band centers shifted away from the Fermi level in the following order: *fcc*-Ni@Pt_{2L} (-1.89 eV), *hcp*-Ni@Pt_{2L}-steps (-2.30 eV), and *hcp*-Ni@Pt_{2L} (-2.32 eV). In the same order, the hydrogen binding energies decreased from -0.46 eV to -0.36 eV and -0.17 eV, respectively. The revised Supplementary **Fig. 22** demonstrates a qualitative correlation between the *d*-band center and the hydrogen binding energy: that is, a downward shift of the *d*-band center corresponds to a reduced hydrogen binding energy, thus improved HER performance based on previous research and our experimental results, which is consistent to the prediction by the *d*-band center theory.

Supplementary Fig. 22 | DFT-calculated relationship between the d -band center and the hydrogen binding energy (ΔG_{H^*}). The d -band centers of $fcc-Ni@Pt_{2L}$, $hcp-Ni@Pt_{2L}$ -steps, and $hcp-Ni@Pt_{2L}$ are -1.89 , -2.30 , and -2.32 eV, respectively. It can be inferred that the trend of the d -band center values is roughly correlated with the trend of the hydrogen binding energy values. According to the d -band center theory, the metastable hcp catalysts weakly adsorb guest molecules, corresponding to low hydrogen binding energies, which promises improved HER performance.

We have discussed the relationship between the d -band center and the hydrogen binding energy (and the consequent HER activity) in the revised manuscript as follows:

Page 22: “Our DFT calculations with the same benchmarks also revealed a rough correlation between the ΔG_{H^*} and the d -band center of the catalysts (Supplementary Fig. 22). A downshift of the d -band center corresponded to a decrease in the ΔG_{H^*} . Based on prior research and our experimental results, the closer proximity of the ΔG_{H^*} to zero is indicative of improved performance in the HER. Therefore, a downshift of the d -band center corresponds to an increase in the HER activity, consistent with the prediction by the d -band center theory.”

Typos:

Page 14 line 13 "efficacy" - efficiency ?

Page 19 line 2 "We further examined, by DFT calculations, the ..." rephrase or omit comma ?

Response: We appreciate the reviewer’s help in correcting the typos.

Overall, given the fact that the material the authors discuss is one Pt/Ni catalyst among many others and open questions concerning the DFT, the stability and characterization of the material, I would recommend to transfer the manuscript to a more specialized journal as it does not match the high standards of a journal like Nature Communications in its current form.

Response: In the response letter and the revised manuscript, the novelty of this work has been thoroughly explained. The density functional theory (DFT) calculations have been strengthened by incorporating new data, and the stability of the material has been thoroughly evaluated through new experiments and characterizations. We sincerely thank the reviewer for the constructive suggestions that improve the manuscript.

Again, we thank both reviewers for their valuable comments and constructive suggestions on our manuscript, which have been greatly helpful for us in improving the quality of this work. We look forward to your kind feedback.

Sincerely yours,

Chuanbo Gao, on behalf of all authors
Xi'an Jiaotong University

REVIEWERS' COMMENTS

Reviewer #3 (Remarks to the Author):

The authors have invested a considerable amount of work in the revision of the manuscript. Especially the addition of further experimental results and some clarification and additional data in the theory section have improved the quality of the work considerably. The detailed and clear reply of the authors to the questions raised by the referees shows that the authors have carefully addressed all points that were raised and their answers are comprehensive and satisfactory, especially as they have introduced the corresponding changes throughout the manuscript, so that the arguments of the authors are also clear to the readers.

Hence, I would now recommend the paper for publication without further revision.